# ColRel: Collaborative Relaying for Federated Learning over Intermittently Connected Networks

**Rajarshi Saha**[1]    **Michal Yemini**[2]    **Emre Ozfatura**[3]
**Deniz Gündüz**[3]    **Andrea J. Goldsmith**[2]
[1]Stanford University    [2]Princeton University    [3]Imperial College London
{rajsaha}@stanford.edu
{myemini, goldsmith}@princeton.edu
{m.ozfatura, d.gunduz}@imperial.ac.uk

## Abstract

Intermittent connectivity of clients to the parameter server (PS) is a major bottleneck in federated edge learning. It induces a large generalization gap, especially when the local data distribution amongst clients exhibits heterogeneity. To overcome communication blockages between clients and the central PS, we have introduced the concept of collaborative relaying (ColRel), wherein the participating clients relay their neighbors' local updates to the PS in order to boost the participation of clients with poor connectivity to the PS. For every communication round, each client initially computes a local consensus of a subset of its neighboring clients' updates and subsequently transmits to the PS, a weighted average of its own update and those of its neighbors'. In this work, we view ColRel as a variance reduction technique that helps in improving the convergence rate for different optimization setups. Consequently, our ColRel approach can be readily integrated as a black box with existing federated learning systems. We provide analytical upper bounds on the resulting convergence rate, which we reduce by optimizing the weights subject to an unbiasedness condition for the global update. Numerical evaluations on the CIFAR-10 dataset demonstrate that our ColRel-based approach achieves a higher test accuracy over Federated Averaging based benchmarks for learning over intermittently-connected networks.

## 1 Introduction

Federated learning (FL) algorithms aim to learn a shared model over data samples that are localized over distributed clients. FL approaches aim to reduce communication overhead and improve clients' privacy by letting each client train a local model on its local dataset and forwarding them periodically to a centralized parameter server (PS). In practical FL scenarios, some clients are stragglers and cannot send their updates regularly. There can be two types of stragglers: (i) *computation stragglers*, which cannot finish their computation within the deadline, or (ii) *communication stragglers*, which cannot transmit their updates to the PS successfully due to communication limitations [7]. The latter may happen when clients suffer from intermittent connectivity to the PS due to temporary blockage of their communication channel [2, 9–11, 31, 47, 55]. In general, persistent stragglers deteriorate the convergence of FL algorithms as the computed local updates become stale and useless, and can even result in bias in the final model. However, communication stragglers that are limited due to loss of direct communication opportunities to the PS are inherently different from computation stragglers, since they can be mitigated by relaying their updates via neighboring clients.

Workshop on Federated Learning: Recent Advances and New Challenges, in Conjunction with NeurIPS 2022 (FL-NeurIPS'22). This workshop does not have official proceedings and this paper is non-archival.

Communication quality at the edge is a key guiding design principle for **FEderated Edge Learning** (FEEL) frameworks [13]. Existing works in the FEEL paradigm have primarily focused on direct communication from the clients to the PS; aimed at improving the performance by resource allocation across clients [3, 4, 8, 13, 14, 28, 33, 36, 38, 39, 46, 49, 57]. However, these works ignore the possibility of cooperation among clients in the case of intermittent communication blockages. Robustness to channel blockages is critical to the reliable operations of mmWave communication systems and robotic systems, where mobile robots explore remote parts of areas of interest and thus can be disconnected from the PS [11, 31, 47, 55].

This work studies the deterioration in performance of existing FL algorithms in the presence of intermittent connections to the PS (Fig. 1). We show that communication stragglers, which suffer from such random unreliable network connectivity, introduce a **topology induced variance (TIV)** in the local updates of clients. To mitigate this, we propose a new FEEL paradigm, which we name Collaborative Relaying (**ColRel**), where client cooperation is utilized to improve connectivity to the PS. In ColRel, clients share their local updates with one another so that each client can send to the PS a weighted average of its own update and those of its neighbors. The PS receives updates from

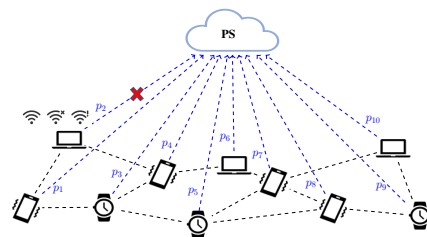

Figure 1: An intermittently connected FL network. Blue and black dotted lines denote intermittent client-PS and client-client connections.

clients with intermittently absent uplink connections, which would otherwise be discarded. ColRel optimizes the averaging weights in order to ensure that the updates received at the PS (i) achieve weak unbiasedness that avoids objective inconsistency, and (ii) minimizes the TIV and subsequently, the convergence time of the learning algorithm under intermittent connectivity.[1]

## 1.1 Significance and Related Works

**Fully Decentralized and Hierarchical FL**: The conventional FL framework [30] is orchestrated by a centralized entity called PS, which helps participating clients reach a consensus on the model parameters by aggregating their locally trained models. Decentralized learning or peer-to-peer FL has been introduced as an alternative, in which the PS is removed to mitigate a potential communication bottleneck and a single point of failure. The aggregation strategy at each client is determined according to the network topology, that is the connection pattern between the clients, and often a fixed topology is considered [17, 18, 21, 26, 37, 44, 45, 51, 54, 56].

An alternative approach to both centralized and decentralized schemes is the *hierarchical FL (HFL)* framework [1, 6, 27, 29], wherein clients are divided into clusters and a PS is assigned to each cluster to perform local aggregation, while the aggregated models at the clusters are later aggregated at the main PS in a subsequent step to obtain the global model. The idea of hierarchical collaborative learning can be redesigned to combine hierarchical and decentralized learning concepts, which is referred to as *semi-decentralized FL*. One of the major challenges in FL that is not considered in the aforementioned works on semi-decentralized FL is the partial client connectivity [12, 48].

The connectivity of the clients is a particularly significant challenge in FEEL. Due to their different physical environments and distances to the PS, clients may have different connectivity to each other and the PS. Customized client selection mechanisms seek a balance between the participation of the clients and the latency for the model aggregation in order to speed up the learning process [4, 15, 28, 35, 36, 40, 46, 49, 50]. In this work, we adopt a different approach to the connectivity problem, and instead of designing a client selection mechanism, or optimizing resource allocation to balance client participation, we introduce a *relaying* mechanism that benefits from the local connectivity between neighboring clients, and ensures that in case of poor connectivity, their local information is conveyed to the PS with the help of their neighbors.

---

[1]A preliminary version of this work was presented in part at the 2022 IEEE International Symposium on Information Theory (ISIT) [52], in which client collaboration was assumed to be over perfectly connected links. An extended version is available in [53], which also considers intermittent client-client links along with local iterations at the clients for smooth and strongly-convex functions.

**Coded Computation for Straggler Mitigation**: Coded computing is a well studied solution for stragglers in distributed learning [23, 24, 32, 34, 43], where data is allocated strategically to clients to create redundant computations that can be exploited by the PS using additional information about the stragglers' identities. Nonetheless, the coded computing paradigm as it is cannot be incorporated into FL systems, which champion the privacy of clients' data as one of their major goals. Our ColRel approach does not require redundancies in clients' data since clients share their *updates* but not their *local data*, and thus preserves privacy to the same extent as conventional FL algorithms. Another notable difference between ColRel and coded computing includes relaying constraints. Coded computation schemes spread the data among clients without constraints on the allowed combinations. In contrast, in our problem setup, clients cannot choose their neighbors as they depend on the channel characteristics, and thus coded computing techniques cannot be readily adopted.

## 1.2 Our Contributions

We start with the problem of distributed mean estimation (DME) in the presence of intermittent connectivity. Our proposed algorithm, ColRel ensures that the mean squared error (MSE) of this estimation is minimized by appropriately relaying the clients' information to the PS. We then study ColRel to mitigate intermittent connectivity for several federated optimization setups. In particular, for an iteration horizon of $T$, we show that: **(i)** For general convex and non-smooth objectives, a subgradient-based ColRel method converges at the rate of $\mathcal{O}\left(\frac{1}{T}\right)$, with constants proportional to the TIV ($\sigma_{\mathrm{tv}}^2$). **(ii)** For smooth functions that satisfy the Polyak-Lojasiewicz condition, gradient descent based ColRel with a constant step-size converges linearly to a neighborhood of the optimal solution, whose size is proportional to $\sigma_{\mathrm{tv}}^2$. **(iii)** For smooth and non-convex objectives, gradient descent based ColRel converges at a rate of $\mathcal{O}\left(\frac{1}{T}\right)$ to the neighborhood of a stationary point (with size proportional to $\sigma_{\mathrm{tv}}^2$). **(iv)** Numerical simulations on FL benchmarks validate our theoretical analyses.

## 2 DME over Intermittently Connected Networks

The DME problem is a fundamental primitive for FL algorithms, where at every iteration, the PS estimates the mean of the local updates of a subset of clients [16, 19, 20, 41, 42]. To this end, analyzing DME is a useful and insightful abstraction. We study DME over intermittently connected networks so as to generalize the conclusion to federated optimization algorithms in subsequent sections. We consider a DME model comprising of $n$ clients, each with a vector $\mathbf{x}_i \in \mathbb{R}^d$, for $i \in [n]$ that satisfy $\|\mathbf{x}_i\| \leq \mathrm{R}$ for some known $\mathrm{R} > 0$, and the goal of the PS is to estimate their mean, i.e., $\overline{\mathbf{x}} \triangleq \frac{1}{n}\sum_{i=1}^n \mathbf{x}_i$. The performance of any estimate $\widehat{\overline{\mathbf{x}}}$ at the PS is measured by its MSE, $\mathcal{E} \triangleq \mathbb{E}\|\widehat{\overline{\mathbf{x}}} - \overline{\mathbf{x}}\|_2^2$.

### 2.1 Communication Model

**Communication between clients and the PS**: We model the intermittent connectivity of any client $i \in [n]$ to the PS by a Bernoulli random variable, $\tau_i \sim \mathrm{Ber}(p_i)$, where $\tau_i = 1$ implies a communication opportunity between client $i$ and the PS, while $\tau_i = 0$ means that the connection is blocked/dropped [12]. We assume that the connectivity is independent across the clients, and the downlink from the PS to the clients is permanently connected.[2] We denote $\mathbf{p} \equiv (p_1, \ldots, p_n)$.

**Communication between clients**: The intermittent connectivity of a transmission from client $i$ to client $j$ is captured by another Bernoulli random variable, $\tau_{ij} \sim \mathrm{Ber}(p_{ij})$, where $p_{ii} = 1$ for every $i \in [n]$. Additionally, if client $j$ can never transmit information successfully to client $i$, we set $p_{ij} = 0$. For simplicity of exposition, we assume that $\tau_{ij}$ and $\tau_{ml}$ are statistically independent for every $i, j, l, m \in [n]$ such that $(i, j) \neq (l, m)$ and $(j, i) \neq (l, m)$. Furthermore, $\tau_{ij}$, and $\tau_l$ are statistically independent for every $i, j, l \in [n]$. We use the notation $\mathrm{E}_{\{i,j\}} = \mathbb{E}[\tau_{ij}\tau_{ji}]$ for every $i, j \in [n]$ to denote the correlation due to channel reciprocity. Finally, we assume that $\mathrm{E}_{\{i,j\}} \geq p_{ij}p_{ji}$, that is, $\mathbb{P}(\tau_{ij} = 1|\tau_{ji} = 1) \geq \mathbb{P}(\tau_{ij} = 1)$, for every $i, j \in [n]$. We use the notation $\mathbf{P} \equiv (p_{ij})_{i,j \in [n]} \in [0, 1]^{n \times n}$.

---

[2]For simplicity, we consider orthogonal communication links from the clients to the PS, where each link is either unavailable or perfect; that is, when it is available it does not suffer from any channel impairments.

**Remark 1.** We assume that the connectivity probabilities $p_i$, $i \in [n]$, are known. In practice, they can be modeled [2] or estimated in a pre-training phase. On the other hand, we do not assume that the instantaneous connectivity information $\{\tau_i, \tau_{ij}\}$ for $i, j \in [n]$, is available to any of the clients. Our prior work [52] considered communication links between pairs of clients (if present) to be perfect, i.e., for any $i \neq j$, $p_{ij} = 0$ or 1, with an extended version [53] that allows for both client-PS and client-to-client communications to be intermittent. This workshop paper generalizes the collaborative approach in these works for the fundamental problem of distributed mean estimation, consequently demonstrating its potential efficacy as a black-box tool that can be integrated into existing FL systems.

## 2.2 ColRel for DME

Since communicating clients can intermittently send their vectors to one another, each client can send the PS a weighted average of its own vector and those of its neighbors. Consequently, the PS can receive the vectors of clients with failing uplink connections. Let $\{\alpha_{ij}\}_{i,j\in[n]}$ denote the collaboration weights. Each client $i$, transmits to the PS the weighted average $\widetilde{\mathbf{x}}_i = \sum_{j\in[n]} \alpha_{ij}\tau_{ji}\mathbf{x}_j$. This transmission is received at the PS with a probability $p_i$. The PS computes the following estimate $\widehat{\widehat{\mathbf{x}}}$ of the mean via ColRel:

$$\widehat{\widehat{\mathbf{x}}} \triangleq \frac{1}{n}\sum_{i\in[n]} \tau_i\widetilde{\mathbf{x}}_i = \frac{1}{n}\sum_{i\in[n]} \tau_i \sum_{j\in[n]} \alpha_{ij}\tau_{ji}\mathbf{x}_j. \tag{1}$$

We denote by $\mathbf{A} \equiv (\alpha_{ij})_{i,j\in[n]}$ the matrix of collaboration weights, where $\alpha_{ij} \geq 0$ for all $i, j \in [n]$.

## 3 TIV: Error Analysis and Optimizing Collaboration

Due to the stochasticity of intermittent connections in the network, the estimate $\widehat{\widehat{\mathbf{x}}}$ is a random variable, and we consider the MSE of $\widehat{\widehat{\mathbf{x}}}$ with respect to the true mean $\overline{\mathbf{x}}$. When $R = 1$, i.e., $\|\mathbf{x}_i\|_2 \leq 1$ for all $i \in [n]$, we refer to the MSE as the **TIV**, $\sigma_{\text{tv}}^2$. Our primary goal is to obtain an unbiased estimate of $\overline{\mathbf{x}}$ at the PS. Under this unbiasedness condition, we derive a worst-case upper bound for $\sigma_{\text{tv}}^2$ in Theorem 3.2. In general, the TIV is a function of connection probabilities and collaboration weights, i.e., $\sigma_{\text{tv}}^2 \equiv \sigma_{\text{tv}}^2(\mathbf{p}, \mathbf{P}, \mathbf{A})$. Consequently, we choose the weights $\mathbf{A} \equiv \{\alpha_{ij}\}_{i,j\in[n]}$ so as to minimize $\sigma_{\text{tv}}^2$. Moreover, we consider $\alpha_{ij} \geq 0$ for all $i, j \in [n]$. In §4 and §5, it is shown how the TIV ($\sigma_{\text{tv}}^2$) affects the convergence performance of federated optimization algorithms.

**Unbiasedness**: Recall that $\alpha_{ji}$ is the collaboration weight client $j$ assigns to the vector it receives from client $i$. Client $i$, and each client $j$ that receives the transmission from client $i$ successfully, i.e. $\tau_{ij} = 1$, try to send to the PS $\alpha_{ji}\mathbf{x}_i$ on behalf of client $i$. Consequently, the accumulated contribution of client $i$ at the PS is given by, $\sum_{j\in[n]} \tau_{ij}\alpha_{ji}\mathbf{x}_i$. The following lemma presents a sufficient condition on the weights $\{\alpha_{ij}\}_{i,j\in[n]}$ that ensures unbiasedness.

**Lemma 3.1 (Sufficient condition for unbiasedness).** *Let $\{\alpha_{ij}\}$ be such that for every $i \in [n]$,*

$$\sum_{j\in[n]} p_j p_{ij} \alpha_{ji} = 1. \tag{2}$$

*Then, $\mathbb{E}\left[\sum_{j\in[n]} \tau_j\tau_{ij}\alpha_{ji}\mathbf{x}_i \mid \mathbf{x}_i\right] = \mathbf{x}_i$, i.e., the contribution of client $i$ at the PS is unbiased.*

As we show next in Theorem 3.2, the effect of the network topology on the TIV is captured by the expression $\mathrm{S}(\mathbf{p}, \mathbf{P}, \mathbf{A})$, which is defined as,

$$\mathrm{S}(\mathbf{p}, \mathbf{P}, \mathbf{A}) \triangleq \sum_{i,j,l\in[n]} p_j(1-p_j)p_{ij}p_{lj}\alpha_{ji}\alpha_{jl} + \sum_{i,j\in[n]} p_{ij}p_j(1-p_{ij})\alpha_{ji}^2 + \sum_{i,l\in[n]} p_ip_l(\mathrm{E}_{\{i,l\}} - p_{il}p_{li})\alpha_{il}\alpha_{li}. \tag{3}$$

**Theorem 3.2.** *For a given $\mathbf{p}, \mathbf{P}$ and $\mathbf{A}$ such that (2) holds, the MSE with ColRel ($\mathcal{E}_{\text{ColRel}}$) satisfies,*

$$\mathcal{E}_{\text{ColRel}}(\mathbf{p}, \mathbf{P}, \mathbf{A}) \leq \frac{\mathrm{R}^2}{n^2} \cdot \mathrm{S}(\mathbf{p}, \mathbf{P}, \mathbf{A}) = \mathrm{R}^2\sigma_{\text{tv}}^2(\mathbf{p}, \mathbf{P}, \mathbf{A}), \quad \text{where} \quad \sigma_{\text{tv}}^2(\mathbf{p}, \mathbf{P}, \mathbf{A}) \triangleq \frac{\mathrm{S}(\mathbf{p}, \mathbf{P}, \mathbf{A})}{n^2}.$$

**Remark 2.** In the absence of any collaboration, we set $p_{ij} = 0$ for all $i \neq j$, which yields suboptimal $\sigma_{\text{tv}}^2$ compared to the weights obtained with optimized collaboration as shown next in §3.1.

### 3.1 Optimizing Collaboration: Minimizing the TIV

As a consequence of Theorem 3.2, we optimize the weight matrix $\mathbf{A}$ so as to minimize $\sigma_{\mathrm{tv}}^2(\mathbf{p}, \mathbf{P}, \mathbf{A})$ subject to the unbiasedness condition (2). In other words, we solve the optimization problem,

$$\min_{\mathbf{A}} \mathrm{S}(\mathbf{p}, \mathbf{P}, \mathbf{A}) \text{ s.t.: } \sum_{j \in [n]} p_j p_{ij} \alpha_{ji} = 1, \ \alpha_{ji} \geq 0 \quad \forall i, j \in [n]. \tag{4}$$

Interestingly, this is the same optimization problem as in [53, Eq. 7]. Due to the term $\sum_{i,l \in [n]} p_i p_l (\mathrm{E}_{\{i,l\}} - p_{il} p_{li}) \alpha_{il} \alpha_{li}$ in (3), problem (4) is not necessarily convex. To this end, as in [53], we minimize a convex relaxation $\overline{\mathrm{S}}(\mathbf{p}, \mathbf{P}, \mathbf{A})$ of $\mathrm{S}(\mathbf{p}, \mathbf{P}, \mathbf{A})$ instead, where,

$$\overline{\mathrm{S}}(\mathbf{p}, \mathbf{P}, \mathbf{A}) \triangleq \sum_{i,j,l \in [n]} p_j (1 - p_j) p_{ij} p_{lj} \alpha_{ji} \alpha_{jl} + \sum_{i,j \in [n]} p_{ij} p_j (1 - p_{ij}) \alpha_{ji}^2 + \sum_{i,l \in [n]} p_i p_l (\mathrm{E}_{\{i,l\}} - p_{il} p_{li}) \alpha_{li}^2. \tag{5}$$

To find the optimal collaboration weights $\mathbf{A}_{\mathrm{opt}}$, we first minimize the convex upper bound $\overline{\mathrm{S}}$ using Gauss-Seidel method. Subsequently, we fine tune this solution by using it as a warm-start initialization for converging to a stationary point of (4). For the sake of completeness, a detailed description of the resulting iterative water-filling type weight optimization algorithm is provided in App. C.

## 4 Federated Optimization over Intermittently Connected Networks

Here, we use the insights from the analysis of DME in §2 and §3 to derive convergence guarantees for federated optimization problems. Our goal is to solve, $\mathbf{x}^* \equiv \arg\min_{\mathbf{x} \in \mathbb{R}^d} \left[ f(\mathbf{x}) = \frac{1}{n} \sum_{i \in [n]} f_i(\mathbf{x}) \right]$, where $\mathbf{x} \in \mathbb{R}^d$ are the parameters of the model, $n$ is the number of clients, $f_i(\mathbf{x})$ is the local loss of model $\mathbf{x}$ on the data stored on client $i$. We consider various standard assumptions of the objective functions $f_i$ and study the convergence guarantees when the gradients computed at the clients are shared with the PS over unreliable networks. We conclude that as a consequence of intermittent connections, the convergence rates are affected by an additional error which is proportional to the TIV $\sigma_{\mathrm{tv}}^2$. The algorithms and their analyses are deferred to the appendices.

### 4.1 Subgradient Method for Non-Smooth and General Convex Objectives

When $f_i$, $i \in [n]$ are convex but not necessarily smooth, the iterates of a *subgradient* method with ColRel in the presence of intermittent connectivity is given by,

$$\mathbf{x}^{t+1} \leftarrow \mathbf{x}^t - \gamma_t \mathbf{g}(\mathbf{x}^t), \quad \text{where} \quad \mathbf{g}(\mathbf{x}^t) = \frac{1}{n} \sum_{i \in [n]} \tau_i^t \sum_{j \in [n]} \alpha_{ij} \tau_{ji}^t \mathbf{g}_j(\mathbf{x}^t). \tag{6}$$

Here, $\mathbf{g}_j(\cdot) \in \partial f_j(\cdot)$ denotes a subgradient of $f_j$, and $\{\tau_i^t, \tau_{ij}^t\}$ are Bernoulli random variables that model the intermittent connectivity at iteration $t$. The convergence of iterations in (6) is stated below.

**Theorem 4.1.** *Suppose that for every $i \in [n]$, the local loss function $f_i$ is $\mathrm{G}_i$ - Lipschitz continuous with $\mathrm{G}_{\max} \triangleq \max_{i \in [n]} \mathrm{G}_i$. Additionally, suppose $f^* \triangleq \inf_{\mathbf{x} \in \mathbb{R}^d} f(\mathbf{x}) > -\infty$ with $f(\mathbf{x}^*) = f^*$, and there exists $\mathrm{R} > 0$ such that $\|\mathbf{x}^0 - \mathbf{x}^*\|_2^2 \leq \mathrm{R}^2$. Then, given $\mathbf{p}$, $\mathbf{P}$, and weights $\mathbf{A}$ satisfying (2), for a learning-rate sequence $\{\gamma_t\}$, we have,*[3]

$$\min_{t=0,\ldots,T-1} \mathbb{E}[f(\mathbf{x}^t)] - f^* \leq \frac{\mathrm{R}^2 + \mathrm{G}_{\max}^2 \left(\sigma_{\mathrm{tv}}^2 + 1\right) \sum_{t=0}^{T-1} \gamma_t^2}{2 \sum_{t=0}^{T-1} \gamma_t}. \tag{7}$$

Theorem 4.1 states that for a constant learning rate choice of $\gamma_t = \gamma$ for all $t = 0, 1, \ldots$, the algorithm (6) converges to a neighborhood of the solution at a rate of $\mathcal{O}(\frac{1}{T})$; with the size of the neighborhood being proportional to the TIV, $\sigma_{\mathrm{tv}}^2$.

---

[3]The assumptions of $f_i$ here are stricter than necessary. We consider them for brevity and they can be relaxed with a more rigorous analysis with minimal variation to the contribution of TIV.

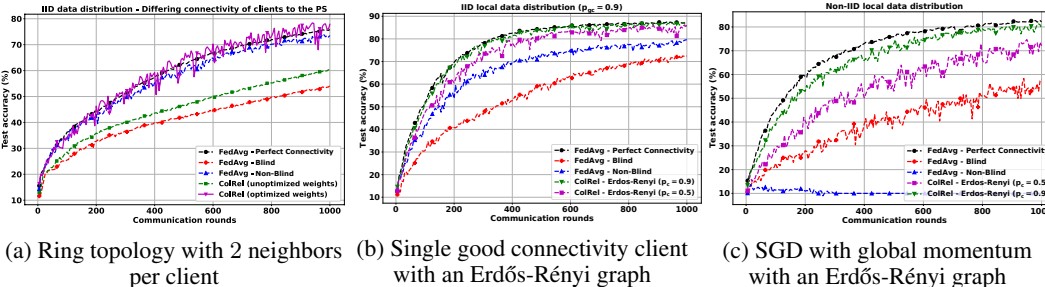

(a) Ring topology with 2 neighbors per client

(b) Single good connectivity client with an Erdős-Rényi graph

(c) SGD with global momentum with an Erdős-Rényi graph

Figure 2: Image classification on the CIFAR-10 dataset using ResNet-20.

## 4.2 Gradient Descent for Smooth and Polyak-Lojasiewicz (PL) Objectives

A function $f$ is said to be $\mu$-PL if $f(\mathbf{x}) - f(\mathbf{x}^*) \leq \frac{1}{2\mu}\|\nabla f(\mathbf{x})\|_2^2$ holds true for all $\mathbf{x} \in \mathbb{R}^d$, where $\mathbf{x}^* \triangleq \arg\min_{\mathbf{x} \in \mathbb{R}^d} f(\mathbf{x})$. Gradient descent with ColRel follows the same iteration as (6) with the *gradient* $\nabla f_j(\mathbf{x}^t)$ instead of a *subgradient* $\mathbf{g}_j(\mathbf{x}^t)$. Theorem 4.2 below states that the gradient descent iterates converge linearly to a neighborhood (with size proportional to $\sigma_{\mathrm{tv}}^2$) of the solution.

**Theorem 4.2.** *Suppose $f$ is $\mu$-PL and $\mathrm{L}$-smooth, and for every $i \in [n]$, $f_i$ is $\mathrm{G}_i$-Lipschitz continuous with $\mathrm{G}_{\max} = \max_{i \in [n]} \mathrm{G}_i$. For a given $\mathbf{p}, \mathbf{P}$, weights $\mathbf{A}$ satisfying (2), and a constant $\gamma \in \left(0, \frac{1}{\mathrm{L}}\right]$, let $\mathbf{x}^{t+1} \leftarrow \mathbf{x}^t - \frac{\gamma}{n} \sum_{i \in [n]} \tau_i^t \sum_{j \in [n]} \alpha_{ij} \tau_{ij}^t \nabla f_j(\mathbf{x}^t)$. Then, we have,*

$$\mathbb{E}[f(\mathbf{x}^T)] - f(\mathbf{x}^*) \leq (1 - \gamma\mu)^T \left(\mathbb{E}[f(\mathbf{x}^0)] - f(\mathbf{x}^*)\right) + \frac{\mathrm{G}_{\max}^2 \sigma_{\mathrm{tv}}^2}{2\mu}. \tag{8}$$

## 4.3 Gradient Descent for Smooth and Non-Convex Objectives

When $f_i$'s are smooth and non-convex, gradient descent with ColRel once again converges to a neighborhood of a stationary point at a rate of $\mathcal{O}(\frac{1}{T})$. This is formalized below.

**Theorem 4.3.** *Suppose $f^* \triangleq \inf_{\mathbf{x} \in \mathbb{R}^d} f(\mathbf{x}) > -\infty$, $f_i, i \in [n]$ is $\mathrm{L}_i$-smooth and $\mathrm{G}_i$-Lipschitz continuous with $\mathrm{G}_{\max} = \max_{i \in [n]} \mathrm{G}_i$, and there exists $\mathrm{D}_0 > 0$ such that $\mathbb{E}f(\mathbf{x}^0) - f^* \leq \mathrm{D}_0$. Then, given $\mathbf{p}, \mathbf{P}$, and weights $\mathbf{A}$ satisfying (2), for $\gamma \in \left(0, \frac{1}{\mathrm{L}}\right]$, we have,*

$$\min_{i=0,\ldots,T-1} \mathbb{E}\left[\|\nabla f(\mathbf{x}^t)\|_2^2\right] \leq \frac{2D_0}{\gamma T} + \mathrm{G}_{\max}^2 \sigma_{\mathrm{tv}}^2. \tag{9}$$

**Remark 3.** To specifically focus on the stochasticity due to random connections, Thms. 4.1, 4.2 and 4.3 state the guarantees with deterministic (sub) gradients at clients. The results can be extended for stochastic (sub) gradients as we show in App. G. We can further improve the convergence rates in these settings by using ColRel together with schemes such as global momentum, periodic averaging, variance reduction, etc. The key takeaway here is that intermittent connectivity over unreliable networks introduces a TIV that can be treated similarly to sources of stochasticity such as stochastic gradient oracle. Furthermore, the TIV can be minimized with optimized ColRel between clients.

## 5 Numerical Simulations

We now provide numerical simulations by training a ResNet-20 model for image classification on the CIFAR-10 dataset [22]. Detailed results and discussions are provided in App. H.

**Optimizing collaboration with heterogeneous client connectivity**: In Fig. 2a, we consider a decentralized ring-topology amongst the clients where $p_{ij} = 1$ if $j = (i - 1, i$ or, $i + 1) \bmod n$, and $p_{ij} = 0$ otherwise. The local datasets of clients are independently and identically distributed (i.i.d.), but they have different connectivity to the PS with $\mathbf{p} = [0.1, 0.2, 0.3, 0.1, 0.1, 0.5, 0.8, 0.1, 0.2, 0.9]$. We observe that with $\{\alpha_{ij}\}$ optimized according to §3.1, ColRel achieves the same performance as with perfect connectivity. We also compare the results with Federated Averaging (FedAvg), where *Blind* refers to when the PS is unaware of the identity of transmitting clients such as in over-the-air

aggregation schemes [3, 4, 33]. Non-Blind refers to when the PS knows exactly how many clients successfully transmitted at each iteration.

**Intermittent client-client connectivity**: In Fig. 2b, a single client has $p_1 = p_{gc} = 0.9$ and $p_i = 0.1$ otherwise. The clients can collaborate intermittently by relaying their update to the good client over an Erdős-Rényi topology with $p_{ij} = p_c$ for $i \neq j$. This resembles a setup where clients are clustered together and only one of them has good connectivity to the PS (for e.g., physical proximity or power constraint). Even with $p_c = 0.5$, the performance is comparable to FedAvg with perfect connectivity.

**Non-IID local data distribution and global momentum**: In Fig. 2c, the CIFAR-10 dataset is sorted and partitioned before distributing it across the clients. Each client has samples from at most 3 classes, in order to emulate a more realistic FL scenario with non-i.i.d. data. There is still just a single client with good PS connectivity as in Fig. 2b. Global momentum is employed at the PS to update the global model. FedAvg (non-blind) fails to converge in this setting, because in the absence of collaboration, clients with important training samples that are critical for training a good model with high accuracy, have a low probability of successful transmission and thus are rarely able to convey their updates to the PS. Consequently, the resulting test accuracy of the global model is $\sim 10\%$, as good as a random classifier for 10 classes. On the other hand, ColRel ensures that the information from these critical datapoints are conveyed to the PS even when the data owner does not have connectivity to the PS.

## 6  Conclusions

FL has been proposed as a distributed learning strategy to train a common model with localized data. However, resource-constrained edge devices often suffer from intermittent connectivity to the PS, and are communication stragglers. On the other hand, thanks to their wireless connectivity, they can communicate locally with their neighbors even when the channel to the PS is blocked. Hence, to mitigate communication stragglers in FL, we proposed a collaborative relaying strategy (ColRel), which exploits the connections between clients to relay potentially missing model updates to the PS. Through the abstraction of DME, we showed that intermittent connectivity introduces a TIV, which we minimized by optimizing the collaboration weights. We then used these optimized weights to improve the convergence rate of FL algorithms. ColRel can be implemented even when the PS is blind to the identities of clients that successfully communicate with it at each round. Numerical results showed the improvement in the training accuracy and the convergence rate that our approach provides in various settings.

## Acknowledgments and Disclosure of Funding

R. Saha, M. Yemini, and A. J. Goldsmith are partially supported by the AFOSR award #002484665 and a Huawei Intelligent Spectrum grant. E. Ozfatura and D. Gündüz received funding from the European Research Council (ERC) through Starting Grant BEACON (no. 677854) and the UK EPSRC (grant no. EP/T023600/1) under the CHIST-ERA program.

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

# Contents

# A Proof of Lemma 3.1: Sufficient Condition for Unbiasedness

Since $\tau_j$ and $\tau_{ij}$ are statistically independent of $\mathbf{x}_i$ and each other, for all $i, j \in [n]$, we have that,

$$\mathbb{E}\left[\sum_{j\in[n]}\tau_j\tau_{ij}\alpha_{ji}\mathbf{x}_i\Big|\mathbf{x}_i\right] = \mathbb{E}\left[\sum_{j\in[n]}\tau_j\tau_{ij}\alpha_{ji}\right]\mathbf{x}_i.$$

Substituting (2) concludes the proof.

Note that the standard model of FL with random client sampling and no connectivity among clients is captured by substituting $p_{ij} = 0, p_i = p, \alpha_{ii} = 1$, and $\alpha_{ij} = 0$ for all $i, j \in [n]$ such that $j \neq i$.

# B Proof of Theorem 3.2: MSE for DME with Client Collaboration under Intermittent Connectivity

The proof of Thm. 3.2 is similar to that of [53, Lemma 6] and utilizes the fact that $\|\mathbf{x}_i\|_2 \leq \mathrm{R}$. Nonetheless, we reproduce it here for the sake of completeness. The MSE for DME with ColRel using a weight matrix $\mathbf{A}$ is given by,

$$\mathcal{E}_{\mathrm{ColRel}}(\mathbf{p}, \mathbf{P}, \mathbf{A}) = \mathbb{E}\|\widehat{\overline{\mathbf{x}}} - \overline{\mathbf{x}}\|_2^2$$

$$= \mathbb{E}\left\|\frac{1}{n}\sum_{i\in[n]}\tau_i\sum_{j\in[n]}\alpha_{ij}\tau_{ji}\mathbf{x}_j - \frac{1}{n}\sum_{i\in[n]}\mathbf{x}_i\right\|_2^2$$

$$= \frac{1}{n^2}\cdot\mathbb{E}\left\|\sum_{i\in[n]}\sum_{j\in[n]}\tau_j\tau_{ij}\alpha_{ji}\mathbf{x}_i - \sum_{i\in[n]}\mathbf{x}_i\right\|_2^2$$

$$= \frac{1}{n^2}\cdot\mathbb{E}\left\|\sum_{i\in[n]}\left(\sum_{j\in[n]}\tau_j\tau_{ij}\alpha_{ji} - 1\right)\mathbf{x}_i\right\|_2^2$$

$$= \frac{1}{n^2}\sum_{i\in[n]}\mathbb{E}\left[\left(\sum_{j\in[n]}\tau_j\tau_{ij}\alpha_{ji} - 1\right)^2\right]\|\mathbf{x}_i\|_2^2$$

$$+ \frac{1}{n^2}\sum_{\substack{i,l\in[n]:\\i\neq l}}\mathbb{E}\left[\left(\sum_{j\in[n]}\tau_j\tau_{ij}\alpha_{ji} - 1\right)\left(\sum_{m\in[n]}\tau_m\tau_{lm}\alpha_{ml} - 1\right)\right]\mathbf{x}_i^\top\mathbf{x}_l. \tag{10}$$

The last equality follows since the random variables $\tau_i \in [n]$ and $\tau_{ij}, i, j \in [n]$ are assumed to be statistically independent of the vectors $\mathbf{x}_i$. Simplifying the first term above gives us,

$$\mathbb{E}\left[\left(\sum_{j\in[n]}\tau_j\tau_{ij}\alpha_{ji} - 1\right)^2\right]$$

$$= \sum_{j\in[n]}\mathbb{E}\left[\tau_j^2\tau_{ij}^2\alpha_{ji}^2\right] + \sum_{j_1,j_2\in[n]:j_1\neq j_2}\mathbb{E}\left[\tau_{j_1}\tau_{j_2}\tau_{ij_1}\tau_{ij_2}\alpha_{j_1i}\alpha_{j_2i}\right] - 2\sum_{j\in[n]}\mathbb{E}\left[\tau_j\tau_{ij}\alpha_{ji}\right] + 1$$

$$\overset{(i)}{=} \sum_{j\in[n]}p_jp_{ij}\alpha_{ji}^2 + \sum_{j_1,j_2\in[n]:j_1\neq j_2}p_{j_1}p_{j_2}p_{ij_1}p_{ij_2}\alpha_{j_1i}\alpha_{j_2i} - 2\sum_{j\in[n]}p_jp_{ij}\alpha_{ji} + 1$$

$$\overset{(ii)}{=} \sum_{j\in[n]}p_jp_{ij}\alpha_{ji}^2 + \left(\sum_{j\in[n]}p_jp_{ij}\alpha_{ji}\right)^2 - \sum_{j\in[n]}p_j^2p_{ij}^2\alpha_{ji}^2 - 1$$

$$\overset{(iii)}{=} \sum_{j \in [n]} p_j p_{ij} \left(1 - p_j p_{ij}\right) \alpha_{ji}^2 \tag{11}$$

Here, (i) follows from the statistical independence of $\tau_i, i \in [n]$ and $\tau_{ij}, i, j \in [n]$, while (ii) and (iii) follow from the sufficient requirement for unbiasedness (2).

To evaluate the second term, note that for every $l \neq i$,

$$\mathbb{E}\left[\left(\sum_{j \in [n]} \tau_j \tau_{ij} \alpha_{ji} - 1\right)\left(\sum_{m \in [n]} \tau_m \tau_{lm} \alpha_{ml} - 1\right)\right]$$

$$= \mathbb{E}\left[\sum_{j \in [n]} \sum_{m \in [n]} \tau_j \tau_m \tau_{ij} \tau_{lm} \alpha_{ji} \alpha_{ml}\right] - \mathbb{E}\left[\sum_{j \in [n]} \tau_j \tau_{ij} \alpha_{ji}\right] - \mathbb{E}\left[\sum_{m \in [n]} \tau_m \tau_{lm} \alpha_{ml}\right] + 1$$

$$= \mathbb{E}\left[\sum_{j \in [n]} \tau_j^2 \tau_{ij} \tau_{lj} \alpha_{ji} \alpha_{jl}\right] + \mathbb{E}\left[\tau_l \tau_i \tau_{il} \tau_{li} \alpha_{li} \alpha_{il}\right] + \mathbb{E}\left[\sum_{m \in [n]: m \neq l, i} \tau_l \tau_m \tau_{il} \tau_{lm} \alpha_{li} \alpha_{ml}\right]$$

$$+ \mathbb{E}\left[\sum_{j \in [n]: j \neq l} \sum_{m \in [n]: m \neq j} \tau_j \tau_m \tau_{ij} \tau_{lm} \alpha_{ji} \alpha_{ml}\right] - 1$$

$$\overset{(i)}{=} \sum_{j \in [n]} p_j p_{ij} p_{lj} \alpha_{ji} \alpha_{jl} + p_i p_l \mathbb{E}_{\{i,l\}} \alpha_{li} \alpha_{il} + \sum_{m \in [n]: m \neq l, i} p_l p_m p_{il} p_{lm} \alpha_{li} \alpha_{ml}$$

$$+ \sum_{j \in [n]: j \neq l} \sum_{m \in [n]: m \neq j} p_j p_m p_{ij} p_{lm} \alpha_{ji} \alpha_{ml} - 1$$

$$\overset{(ii)}{=} \sum_{j \in [n]} p_j (1 - p_j) p_{ij} p_{lj} \alpha_{ji} \alpha_{jl} + p_i p_l (\mathbb{E}_{\{i,l\}} - p_{il} p_{li}) \alpha_{il} \alpha_{li}. \tag{12}$$

Here, (i) follows from the statistical independence of $\tau$'s as before. The equality (ii) follows from the sufficient conditions for unbiasedness which lead to the following chain of equalities:

$$\sum_{m \in [n]: m \neq l, i} p_l p_m p_{il} p_{lm} \alpha_{li} \alpha_{ml} + \sum_{j \in [n]: j \neq l} \sum_{m \in [n]: m \neq j} p_j p_m p_{ij} p_{lm} \alpha_{ji} \alpha_{ml}$$

$$= p_l p_{il} \alpha_{li} \sum_{m \in [n]: m \neq l, i} p_m p_{lm} \alpha_{ml} + \sum_{j \in [n]: j \neq l} p_j p_{ij} \alpha_{ji} \sum_{m \in [n]: m \neq j} p_m p_{lm} \alpha_{ml}$$

$$= p_l p_{il} \alpha_{li} (1 - p_i p_{li} \alpha_{il} - p_l p_{ll} \alpha_{ll}) + \sum_{j \in [n]: j \neq l} p_j p_{ij} \alpha_{ji} (1 - p_j p_{lj} \alpha_{jl})$$

$$= p_l p_{il} \alpha_{li} (1 - p_i p_{li} \alpha_{il} - p_l p_{ll} \alpha_{ll}) + \sum_{j \in [n]: j \neq l} p_j p_{ij} \alpha_{ji} - \sum_{j \in [n]: j \neq l} p_j^2 p_{ij} p_{lj} \alpha_{ji} \alpha_{jl}$$

$$= p_l p_{il} \alpha_{li} (1 - p_i p_{li} \alpha_{il} - p_l p_{ll} \alpha_{ll}) + 1 - p_l p_{il} \alpha_{li} - \sum_{j \in [n]: j \neq l} p_j^2 p_{ij} p_{lj} \alpha_{ji} \alpha_{jl}$$

$$= -p_l p_{il} \alpha_{li} (p_i p_{li} \alpha_{il} + p_l p_{ll} \alpha_{ll}) + 1 - \sum_{j \in [n]: j \neq l} p_j^2 p_{ij} p_{lj} \alpha_{ji} \alpha_{jl}$$

$$= -p_l p_{il} \alpha_{li} (p_i p_{li} \alpha_{il} + p_l p_{ll} \alpha_{ll}) + 1 - \sum_{j \in [n]} p_j^2 p_{ij} p_{lj} \alpha_{ji} \alpha_{jl} + p_l^2 p_{il} p_{ll} \alpha_{li} \alpha_{ll}$$

$$= -p_i p_l p_{il} p_{li} \alpha_{il} \alpha_{li} + 1 - \sum_{j \in [n]} p_j^2 p_{ij} p_{lj} \alpha_{ji} \alpha_{jl}. \tag{13}$$

From (11) and (12), the MSE with ColRel in (10) simplifies to,

$$\mathcal{E}_{\text{ColRel}}(\mathbf{p}, \mathbf{P}, \mathbf{A}) = \frac{1}{n^2} \sum_{i,j \in [n]} p_j p_{ij} (1 - p_j p_{ij}) \alpha_{ji}^2 \cdot \|\mathbf{x}_i\|_2^2$$

$$+ \frac{1}{n^2} \sum_{\substack{i,l\in[n]:\\ i\neq l}} \left( \sum_{j\in[n]} p_j(1-p_j)p_{ij}p_{lj}\alpha_{ji}\alpha_{jl} + p_i p_l (\mathrm{E}_{\{i,l\}} - p_{il}p_{li})\alpha_{il}\alpha_{li} \right) \cdot \mathbf{x}_i^\top \mathbf{x}_l$$

$$= \frac{1}{n^2} \sum_{i,j,l\in[n]} p_j(1-p_j)p_{ij}p_{lj}\alpha_{ji}\alpha_{jl} \cdot \mathbf{x}_i^\top \mathbf{x}_l$$

$$+ \frac{1}{n^2} \sum_{i,j\in[n]} \left[ p_j p_{ij}(1-p_j p_{ij}) - p_j(1-p_j)p_{ij}^2 \right] \alpha_{ji}^2 \cdot \|\mathbf{x}_i\|_2^2$$

$$+ \frac{1}{n^2} \sum_{\substack{i,l\in[n]:\\ i\neq l}} p_i p_l \left( \mathrm{E}_{\{i,l\}} - p_{il}p_{li} \right) \alpha_{il}\alpha_{li} \cdot \mathbf{x}_i^\top \mathbf{x}_l$$

$$= \frac{1}{n^2} \sum_{i,j,l\in[n]} p_j(1-p_j)p_{ij}p_{lj}\alpha_{ji}\alpha_{jl} \cdot \mathbf{x}_i^\top \mathbf{x}_l$$

$$+ \frac{1}{n^2} \sum_{i,j\in[n]} p_{ij}p_j(1-p_{ij})\alpha_{ji}^2 \cdot \|\mathbf{x}_i\|_2^2$$

$$+ \frac{1}{n^2} \sum_{i,l\in[n]} p_i p_l \left( \mathrm{E}_{\{i,l\}} - p_{il}p_{li} \right) \alpha_{il}\alpha_{li} \cdot \mathbf{x}_i^\top \mathbf{x}_l, \tag{14}$$

where the last inequality follows from $\mathrm{E}_{\{i,i\}} - p_{ii}p_{ii} = 1 - 1\cdot 1 = 0$.

Note that we initially assumed $\mathrm{E}_{\{i,l\}} \geq p_{il}p_{li}$, and constrained our weights $\alpha_{il}, \alpha_{li}$ to be non-negative. Furthermore, since $\|\mathbf{x}_i\|_2 \leq \mathrm{R}, \forall\, i \in [n]$, an application of Cauchy-Schwarz inequality to upper bound the inner products yields, $\mathbf{x}_i^\top \mathbf{x}_l \leq \|\mathbf{x}_i\|_2\|\mathbf{x}_l\|_2 \leq \mathrm{R}^2$. This yields,

$$\mathcal{E}_{\mathrm{ColRel}}(\mathbf{p}, \mathbf{P}, \mathbf{A}) \leq \frac{\mathrm{R}^2}{n^2} \mathrm{S}(\mathbf{p}, \mathbf{P}, \mathbf{A}), \tag{15}$$

where, $\mathrm{S}(\mathbf{p}, \mathbf{P}, \mathbf{A})$ is defined as in (3). This completes the proof.

## C  Optimizing the Collaboration Weights: Minimizing $\sigma_{\mathrm{tv}}^2(\mathbf{p}, \mathbf{P}, \mathbf{A})$

In this section, we show how to choose the optimal weights for collaboration. The results of this section follow from [53]. We omit the proofs and refer the reader to [53, Lemma 2] and [53, Lemma 7]. As discussed already in §3 and §4, in order to reduce the TIV, we want to solve the following optimization problem (4) (reproduced below):

$$\min_{\mathbf{A}} \mathrm{S}(\mathbf{p}, \mathbf{P}, \mathbf{A}) \text{ s.t.:} \sum_{j\in[n]} p_j p_{ij}\alpha_{ji} = 1,\ \alpha_{ji} \geq 0 \quad \forall i,j \in [n]. \tag{16}$$

Here, $\mathrm{S}(\mathbf{p}, \mathbf{P}, \mathbf{A})$ is given by,

$$\mathrm{S}(\mathbf{p}, \mathbf{P}, \mathbf{A}) \triangleq \sum_{i,j,l\in[n]} p_j(1-p_j)p_{ij}p_{lj}\alpha_{ji}\alpha_{jl} + \sum_{i,j\in[n]} p_{ij}p_j(1-p_{ij})\alpha_{ji}^2 + \sum_{i,l\in[n]} p_i p_l (\mathrm{E}_{\{i,l\}} - p_{il}p_{li})\alpha_{il}\alpha_{li}. \tag{17}$$

### C.1  Deriving the Convex Relaxation for Minimizing $\mathrm{S}(\mathbf{p}, \mathbf{P}, \mathbf{A})$

Due to the last term $\sum_{i,l\in[n]} p_i p_l (\mathrm{E}_{\{i,l\}} - p_{il}p_{li})\alpha_{il}\alpha_{li}$, the function $\mathrm{S}(\mathbf{p}, \mathbf{P}, \mathbf{A})$ is not necessarily convex with respect to $\mathbf{A}$. To this end, we consider the convex upper bound $\overline{\mathrm{S}}(\mathbf{p}, \mathbf{P}, \mathbf{A})$ where,

$$\overline{\mathrm{S}}(\mathbf{p}, \mathbf{P}, \mathbf{A}) \triangleq \sum_{i,j,l\in[n]} p_j(1-p_j)p_{ij}p_{lj}\alpha_{ji}\alpha_{jl} + \sum_{i,j\in[n]} p_{ij}p_j(1-p_{ij})\alpha_{ji}^2 + \sum_{i,l\in[n]} p_i p_l (\mathrm{E}_{\{i,l\}} - p_{il}p_{li})\alpha_{li}^2. \tag{18}$$

The following lemma formally proves the validity of this convex relaxation.

**Lemma C.1.** *[53, Lemma 2] For every* $\mathbf{A}$ *such that* $\alpha_{ij} \geq 0$, $\forall\, i,j \in [n]$, $\mathrm{S}(\mathbf{p},\mathbf{P},\mathbf{A}) \leq \overline{\mathrm{S}}(\mathbf{p},\mathbf{P},\mathbf{A})$. *Furthermore,* $\overline{\mathrm{S}}(\mathbf{p},\mathbf{P},\mathbf{A})$ *is convex with respect* $\mathbf{A}$ *for* $\mathbf{p} \in [0,1]^n$.

The proof of Lemma C.1 utilizes the result from the following auxiliary lemma.

**Lemma C.2.** *[53, Lemma 7] Let* $\mathbf{y}, \mathbf{c} \in \mathbb{R}^{\tilde{d}}$ *where* $\tilde{d} \in \mathbb{N}_+$. *Denote* $h_{\mathbf{c}}(\mathbf{y}) = \left( \sum_{i=1}^{\tilde{d}} \mathbf{c}_i \mathbf{y}_i \right)^2$, *then* $h_{\mathbf{c}}(\mathbf{y})$ *is convex.*

Lemmas C.1 implies that the following optimization problem is a convex relaxation of (4):

$$\min_{\mathbf{A}} \overline{\mathrm{S}}(\mathbf{p},\mathbf{P},\mathbf{A}) \ \text{s.t.:} \ \sum_{j\in[n]} p_j p_{ij} \alpha_{ji} = 1, \ \alpha_{ji} \geq 0 \quad \forall i,j \in [n]. \tag{19}$$

## C.2   Algorithm to optimize the collaboration weights

Let $\mathbf{A}_i$ denote the $i^{\text{th}}$ column of $\mathbf{A}$, that is, $\mathbf{A}_i \triangleq (\mathbf{A}_{1i}, \ldots, \mathbf{A}_{ni})^\top$. Since the domain of the problems (4) and (19) is separable with respect to $\mathbf{A}_i$, we can use the Gauss-Seidel method to iteratively solve (19) and converge to an optimal solution [5, Proposition 2.7.1] for the upper bound (19). We can then converge to a stationary point of (4) in the vicinity of this solution by employing the Gauss-Seidel method once again.

**Remark 1.** We note that when $p_{ij} \in \{0,1\}$, $\forall i,j \in [n]$, i.e., the client-client connections are always present or always absent, the problem (4) is convex. Additionally, in this case the problems (4) and (19) coincide since $\mathrm{E}_{\{i,l\}} - p_{il}p_{li} = 0$ for every $i,j \in [n]$.

**Optimizing the convex relaxation** (19): Let $\mathbf{A}_i^{(\ell)}$ denote the approximated value for $\mathbf{A}_i$ in the $\ell^{\text{th}}$ iteration of the Gauss-Seidel method of (19). We choose the initial solution to be

$$\mathbf{A}_{ji}^{(0)} = \frac{1}{\left( \sum_{k\in[n]} \mathbb{1}_{\{p_k>0, p_{ik}>0\}} \right)} p_j p_{ij} \cdot \mathbb{1}_{\{p_j>0, p_{ij}>0\}}.$$

We can improve our solution iteratively by the Gauss-Seidel method until convergence to an optimal point of (19). That is, at every iteration $\ell$ we compute $\mathbf{A}^{(\ell)}$ as follows,

$$\mathbf{A}_i^{(\ell)} = \begin{cases} \widehat{\mathbf{A}}_i^{(\ell)} & \text{if } \ell \mod n + n \cdot \mathbb{1}_{\{\ell \mod n=0\}} = i \\ \mathbf{A}_i^{(\ell-1)} & \text{otherwise} \end{cases} \tag{20}$$

where,

$$\widehat{\mathbf{A}}_i^{(\ell)} = \arg\min \left[ \sum_{j\in[n]} p_j p_{ij} \left(1 - p_j p_{ij}\right) \alpha_{ji}^2 + 2 \sum_{l\in[n]:l\neq i} \sum_{j\in[n]} p_j (1-p_j) p_{ij} p_{lj} \alpha_{ji} \alpha_{jl}^{(\ell-1)} \right.$$

$$\left. + \sum_{j\in[n]} p_i p_j (\mathrm{E}_{\{i,j\}} - p_{ij} p_{ji}) \alpha_{ji}^2 \right],$$

$$\text{s.t.:} \sum_{j\in[n]} p_j p_{ij} \alpha_{ji} = 1, \ \alpha_{ji} \geq 0, \quad \forall j \in [n]. \tag{21}$$

Using Lagrange multipliers, we show in App. C.3 that the optimal value for $\mathbf{A}_i^{(\ell)}$ is:

$$\widehat{\alpha}_{ji}^{(\ell)}(\lambda_i) = \begin{cases} \left( \dfrac{-2(1-p_j)\sum_{l\in[n]:l\neq i} p_{lj}\alpha_{jl}^{(\ell-1)} + \lambda_i}{2[(1-p_j p_{ij}) + p_i(\mathrm{E}_{\{i,j\}}/p_{ij} - p_{ji})]} \right)^+ & \text{if } p_j p_{ij} \cdot \max_{k\in[n]} p_k p_{ik} \in (0,1), \\ \dfrac{1}{\sum_{k\in[n]} \mathbb{1}_{\{p_k p_{ik}=1\}}} & \text{if } p_j p_{ij} = 1, \\ 0 & \text{otherwise} \end{cases} \tag{22}$$

where $(a)^+ \triangleq \max\{a, 0\}$ and $\lambda_i$ is set such that $\sum_{j\in[n]} p_j p_{ij} \widehat{\alpha}_{ji}^{(\ell)}(\lambda_i) = 1$. We can find $\lambda_i$ using the bisection method over the interval:

$$\left[ 0, \max_{j:p_j p_{ij}\in(0,1)} \left\{ \frac{2[(1-p_j p_{ij}) + p_i(\mathrm{E}_{\{i,j\}}/p_{ij} - p_{ji})]}{p_j p_{ij}} + 2(1-p_j) \sum_{l\in[n]:l\neq i} p_{lj}\alpha_{jl}^{(\ell-1)} \right\} \right]$$

**Fine tuning** (4): Let $\mathbf{A}_i^{(\ell)}$ denote the approximated value for $\mathbf{A}_i$ in the $\ell^{\text{th}}$ iteration of the Gauss-Seidel method of (4), and assume a given initialization $\mathbf{A}_i^{(0)}$. We can improve our solution iteratively by the Gauss-Seidel method until convergence to a stationary point of (4).

At every iteration $\ell$ we compute $\mathbf{A}_i^{(\ell)}$ as follows,

$$\mathbf{A}_i^{(\ell)} = \begin{cases} \widehat{\mathbf{A}}_i^{(\ell)} & \text{if } \ell \mod n + n \cdot \mathbb{1}_{\{\ell \mod n = 0\}} = i, \\ \mathbf{A}_i^{(\ell-1)} & \text{otherwise,} \end{cases} \tag{23}$$

where $\mathbb{1}_{\{\cdot\}}$ denotes the indicator function, and

$$\widehat{\mathbf{A}}_i^{(\ell)} = \arg\min \left[ \sum_{j \in [n]} p_j p_{ij} \left(1 - p_j p_{ij}\right) \alpha_{ji}^2 + 2 \sum_{l \in [n]: l \neq i} \sum_{j \in [n]} p_j (1 - p_j) p_{ij} p_{lj} \alpha_{ji} \alpha_{jl}^{(\ell-1)} \right.$$
$$\left. + 2 \sum_{j \in [n]} p_i p_j (\mathrm{E}_{\{i,j\}} - p_{ij} p_{ji}) \alpha_{ij} \alpha_{ji} \right],$$

$$\text{s.t.: } \sum_{j \in [n]} p_j p_{ij} \alpha_{ji} = 1, \quad \alpha_{ji} \geq 0, \quad \forall j \in [n]. \tag{24}$$

Using Lagrange multipliers, we show in Appendix C.3 that the optimal value for $\widehat{\mathbf{A}}_i^{(\ell)}$ is as follows:

$$\widehat{\alpha}_{ji}^{(\ell)}(\lambda_i) = \begin{cases} \left( \dfrac{-2(1-p_j) \sum\limits_{l \in [n]: l \neq i} p_{lj} \alpha_{jl}^{(\ell-1)} - 2p_i \left( \frac{\mathrm{E}_{\{i,j\}}}{p_{ij}} - p_{ji} \right) \alpha_{ij}^{(\ell-1)} + \lambda_i}{2(1 - p_j p_{ij})} \right)^+ & \text{if } p_j p_{ij} \max\limits_{k \in [n]} p_k p_{ik} \in (0,1), \\ \dfrac{1}{\sum_{k \in [n]} \mathbb{1}_{\{p_k p_{ik} = 1\}}} & \text{if } p_j p_{ij} = 1, \\ 0 & \text{otherwise} \end{cases},$$
$$\tag{25}$$

where $\lambda_i$ is set such that $\sum_{j \in [n]} p_j p_{ij} \widehat{\alpha}_{ji}^{(\ell)}(\lambda_i) = 1$. We can find $\lambda_i$ using the bisection method over the interval:

$$\left[ 0, \max_{j: p_j p_{ij} \in (0,1)} \left\{ \frac{2(1 - p_j p_{ij})}{p_j p_{ij}} + 2(1 - p_j) \sum_{l \in [n]: l \neq i} p_{lj} \alpha_{jl}^{(\ell-1)} + 2p_i \left( \frac{\mathrm{E}_{\{i,j\}}}{p_{ij}} - p_{ji} \right) \alpha_{ij}^{(\ell-1)} \right\} \right].$$

We summarize the centralized optimization procedure for $\mathbf{A}$ in Algorithm 1.

When the connectivity between two communicating clients is reliable, i.e., two clients are either connected with high probability or disconnected, Algorithm 1 can be implemented in a distributed fashion by the clients. In this case each client does not need to fully know $\mathbf{p}, \mathbf{P}$ and $\mathbf{A}$, but only the weights and transmission probabilities of all its direct neighbors and its second degree neighbors (i.e., neighbors of its neighbors). Such a distributed algorithm can be used to optimize the weights when the PS is blind to the identities of the transmitting clients at all times and cannot use a training period for learning the transmission probabilities. Finally, we note the weights resulting from Algorithm 1 can be used as long as the probabilities $\mathbf{p}$ and $\mathbf{P}$ are fixed, and are not needed to be calculated in every communication round.

**Computation complexity** The overall computational complexity of Algorithm 1 is $O(I \cdot (n^2 + K))$, where $K$ is the number of iterations used in the bisection method for optimizing $\lambda_i$.

### C.3 Iterative Gauss-Seidel update using Lagrange multipliers

First, observe that we can set $\alpha_{ji} = 0$ for every $j$ such that $p_{ij} p_j = 0$. Additionally, if $\max_{k \in [n]} \{p_k p_{ik}\} = 1$, then we can set $\alpha_{ji} = \mathbb{1}_{\{p_{ij} p_j = 1\}} \cdot \left( \sum_{k \in [n]} \mathbb{1}_{\{p_k p_{ik} = 1\}} \right)^{-1}$. Therefore, hereafter we assume that $i$ is such that $\max_{k \in [n]} \{p_k p_{ik}\} < 1$ and that $j$ is such that $p_{ij} p_j \in (0,1)$. We proceed to solve each of the problems (21) and (24), respectively.

**Solving the Convex Optimization Problem** (21): The Lagrangian of (21) is,

**Algorithm 1:** COPT-$\mathbf{A}$ Centralized optimization of the weight matrix $\mathbf{A}$

---

**Input:** A set of clients $[n]$, the functions $\overline{\mathrm{S}}(\mathbf{p}, \mathbf{P}, \mathbf{A})$ and $\mathrm{S}(\mathbf{p}, \mathbf{P}, \mathbf{A})$, vector of connectivity probabilities $\mathbf{p}$, matrix of inter-client connectivity probabilities $\mathbf{P}$, maximal number of iteration $I$.

**Output:** A matrix $\mathbf{A}^{(L)}$ that approximately minimizes (4)

1 **Initialize (convex relaxation):** $\mathbf{A}_{ji}^{(0)} = \frac{1}{\left(\sum_{j \in [n]} \mathbb{1}_{\{p_j > 0, p_{ij} > 0\}}\right) p_j p_{ij}} \cdot \mathbb{1}_{\{p_j > 0, p_{ij} > 0\}}$, and $\ell = 0$.

2 **while** $\ell \leq I - 1$ **do**

3 $\quad$ $\ell \leftarrow \ell + 1$

4 $\quad$ $i \leftarrow \ell \mod n + n \cdot \mathbb{1}_{\{\ell \mod n = 0\}}$

5 $\quad$ Compute $\widehat{\mathbf{A}}_i^{(\ell)}$ according to (22)

6 $\quad$ Set $\mathbf{A}_k^{(\ell)}$ according to (20) for every $k \in [n]$

7 **end**

8 **Warm-initialize (fine tuning):** $\mathbf{A}^{(0)} = \mathbf{A}^{(L)}$ and $\ell = 0$.

9 **while** $\ell \leq I - 1$ **do**

10 $\quad$ $\ell \leftarrow \ell + 1$

11 $\quad$ $i \leftarrow \ell \mod n + n \cdot \mathbb{1}_{\{\ell \mod n = 0\}}$

12 $\quad$ Compute $\widehat{\mathbf{A}}_i^{(\ell)}$ according to (25)

13 $\quad$ Set $\mathbf{A}^{(\ell)}$ according to (23) for every $k \in [n]$

14 **end**

---

$$\overline{\mathcal{L}}(\mathbf{A}_i^{(\ell)}, \lambda_i) = \sum_{j \in [n]} p_j p_{ij} (1 - p_j p_{ij}) \alpha_{ji}^2 + 2 \sum_{l \in [n]:l \neq i} \sum_{j \in [n]} p_j (1 - p_j) p_{ij} p_{lj} \alpha_{ji} \alpha_{jl}^{(\ell-1)}$$

$$+ \sum_{j \in [n]} p_i p_j (\mathrm{E}_{\{i,j\}} - p_{ij} p_{ji}) \alpha_{ji}^2 - \lambda_i \left( \sum_{j \in [n]} p_j p_{ij} \alpha_{ji} - 1 \right) - \mu_{ji}(\alpha_{ji}).$$

Additionally,

$$\frac{\partial \overline{\mathcal{L}}(\mathbf{A}_i^{(\ell)}, \lambda_i)}{\partial \alpha_{ji}} = 2p_j[p_{ij}(1 - p_j p_{ij}) + p_i(\mathrm{E}_{\{i,j\}} - p_{ij} p_{ji})]\alpha_{ji}$$

$$+ 2p_j(1 - p_j)p_{ij} \sum_{l \in [n]:l \neq i} p_{lj} \alpha_{jl}^{(\ell-1)} - \lambda_i p_{ij} p_j + \mu_{ji},$$

$$\frac{\partial \overline{\mathcal{L}}(\mathbf{A}_i^{(\ell)}, \lambda_i)}{\partial \lambda_i} = 1 - \sum_{j \in [n]} p_j p_{ij} \alpha_{ji}, \qquad \frac{\partial \overline{\mathcal{L}}(\mathbf{A}_i^{(\ell)}, \lambda_i)}{\partial \mu_{ji}} = -\alpha_{ji}.$$

It follows from the Karush–Kuhn–Tucker conditions that

$$\alpha_{ji}(\lambda_i) = \left( \frac{-2(1 - p_j) \sum_{l \in [n]:l \neq i} p_{lj} \alpha_{jl}^{(\ell-1)} + \lambda_i}{2[(1 - p_j p_{ij}) + p_i(\mathrm{E}_{\{i,j\}}/p_{ij} - p_{ji})]} \right)^+,$$

and $\lambda_i \geq 0$ is set such that $\sum_{j \in [n]} p_j p_{ij} \alpha_{ji}(\lambda_i) = 1$.

**Solving the Convex Optimization Problem** (24): The Lagrangian of (24) is

$$\mathcal{L}(\mathbf{A}_i^{(\ell)}, \lambda_i) = \sum_{j \in [n]} p_j p_{ij} (1 - p_j p_{ij}) \alpha_{ji}^2 + 2 \sum_{l \in [n]:l \neq i} \sum_{j \in [n]} p_j (1 - p_j) p_{ij} p_{lj} \alpha_{ji} \alpha_{jl}^{(\ell-1)}$$

$$+ 2 \sum_{j \in [n]} p_i p_j (\mathrm{E}_{\{i,j\}} - p_{ij} p_{ji}) \alpha_{ji} \alpha_{ij}^{(\ell-1)} - \lambda_i \left( \sum_{j \in [n]} p_j p_{ij} \alpha_{ji} - 1 \right) - \mu_{ji}(\alpha_{ji}).$$

Additionally,

$$\frac{\partial \mathcal{L}(\mathbf{A}_i^{(\ell)}, \lambda_i)}{\partial \alpha_{ji}} = 2p_j p_{ij}\left(1 - p_j p_{ij}\right) + 2p_j(1 - p_j)p_{ij} \sum_{l \in [n]: l \neq i} p_{lj}\alpha_{jl}^{(\ell-1)}$$

$$+ 2p_i p_j (\mathrm{E}_{\{i,j\}} - p_{ij}p_{ji})\alpha_{ij}^{(\ell-1)} - \lambda_i p_{ij}p_j + \mu_{ji},$$

$$\frac{\partial \mathcal{L}(\mathbf{A}_i^{(\ell)}, \lambda_i)}{\partial \lambda_i} = 1 - \sum_{j \in [n]} p_j p_{ij}\alpha_{ji}, \qquad \frac{\partial \mathcal{L}(\mathbf{A}_i^{(\ell)}, \lambda_i)}{\partial \mu_{ji}} = -\alpha_{ji}.$$

It follows from the Karush–Kuhn–Tucker conditions that

$$\alpha_{ji}(\lambda_i) = \left( \frac{-2(1 - p_j)\sum_{l \in [n]: l \neq i} p_{lj}\alpha_{jl}^{(\ell-1)} - 2p_i(\mathrm{E}_{\{i,j\}}/p_{ij} - p_{ji})\alpha_{ij}^{(\ell-1)} + \lambda_i}{2\left(1 - p_j p_{ij}\right)} \right)^+,$$

and $\lambda_i \geq 0$ is set such that $\sum_{j \in [n]} p_j p_{ij}\alpha_{ji}(\lambda_i) = 1$.

## D  Proof of Theorem 4.1: Non-Smooth and General Convex Objectives

In this section, we discuss and analyze in detail, a subgradient method with collaborative relaying for federated optimization over intermittently connected networks. The convergence rate of this analysis is derived following the analysis of DME and we use the optimized collaboration weights, i.e., $\mathbf{A} = \mathbf{A}_{\mathrm{opt}}$ in order to minimize the TIV, $\sigma_{\mathrm{tv}}^2(\mathbf{p}, \mathbf{P}, \mathbf{A})$. The algorithm pseudocode is presented in Alg. 2.

---

Algorithm 2: **ColRel-Subgradient**: Subgradient method with ColRel

---

**Input:** A set of clients $[n]$ with local losses $f_i$, a client connectivity graph G, optimized weight matrix $\mathbf{A} = \mathbf{A}_{\mathrm{opt}}$, maximal number of iteration $T$, initial point $\mathbf{x}^0$, learning-rate sequence $\{\gamma_t\}_{t=0,\ldots,T-1}$.
**Output:** An approximate minimizer $\mathbf{x}^*$ of $f(\mathbf{x}) = \frac{1}{n}\sum_{i \in [n]} f_i(\mathbf{x})$.

1 **Initialize**: Iteration index $t = 0$.
2 **while** $t < T$ **do**
3     **PS**: Broadcasts $\mathbf{x}^t$ to all clients $i \in [n]$.
4     **for** $i \in [n]$ **do**
5         (All clients execute in parallel)
6         Client-$i$ computes a subgradient $\mathbf{g}_i(\mathbf{x}^t) \in \partial f_i(\mathbf{x}^t)$.
7         **Local collaboration**: Client-$i$ broadcasts $\mathbf{g}_i(\mathbf{x}^t)$ to all (neighboring) nodes, while also receiving $\mathbf{g}_j(\mathbf{x}^t), j \in [n], j \neq i$ from them over intermittently connected links.
8         **Local consensus**: Client-$i$ computes a weighted average of updates received
                $\widetilde{\mathbf{g}}_i(\mathbf{x}^t) = \sum_{j \in [n]} \alpha_{ij}\tau_{ij}^t \mathbf{g}_j(\mathbf{x}^t)$.
9         Client-$i$ sends $\widetilde{\mathbf{g}}_i(\mathbf{x}^t)$ to the PS over an intermittently connected link.
10     **end**
11     **PS**: Receives $\tau_i^t \widetilde{\mathbf{g}}_i(\mathbf{x}^t)$ from all clients $i \in [n]$.
12     **PS**: Computes global subgradient by (blindly) aggregating all received updates,
        $\mathbf{g}_i(\mathbf{x}^t) = \frac{1}{n}\sum_{i \in [n]} \tau_i^t \widetilde{\mathbf{g}}_i(\mathbf{x}^t)$.
13     **PS**: Takes a subgradient step: $\mathbf{x}^{t+1} \leftarrow \mathbf{x}^t - \gamma_t \mathbf{g}(\mathbf{x}^t)$.
14     (Increment iteration counter) $t \leftarrow t + 1$.
15 **end**

---

To derive the convergence rate of Alg. 2 for non-smooth and general convex objective functions, we make the following assumptions (as stated in statement of Theorem 4.3).

**Assumption D.1.** For $i \in [n]$, the local loss function $f_i$ is $\mathrm{G}_i$-Lipschitz continuous. That is, for any $\mathbf{x}, \mathbf{y} \in \mathbb{R}^d$, $|f_i(\mathbf{x}) - f_i(\mathbf{y})| \leq \mathrm{G}_i\|\mathbf{x} - \mathbf{y}\|_2$.

$G_i$-Lipschitz continuity implies for any $\mathbf{x} \in \mathbb{R}^d$, the subgradient $\mathbf{g}_i(\mathbf{x}) \in \partial f_i(\mathbf{x})$ satisfies $\|\mathbf{g}_i(\mathbf{x})\|_2 \leq G_i$. Here, $\partial f_i(\mathbf{x})$ denotes the subdifferential of $f_i$ at $\mathbf{x}$.

**Assumption D.2.** The objective function is lower bounded and the infimum is attainable, i.e.,

$$f^* = \inf_{\mathbf{x} \in \mathbb{R}^d} f(\mathbf{x}) > -\infty, \quad \text{with } f(\mathbf{x}^*) = f^*. \tag{26}$$

**Assumption D.3.** The initial distance to optimality can be upper bounded by some known $R > 0$, i.e.,

$$\|\mathbf{x}^0 - \mathbf{x}^*\|_2^2 \leq R^2. \tag{27}$$

In general, these assumptions can be relaxed. Here, they are merely made here to simplify the proofs. Denote, $G_{\max} \triangleq \max_{i \in [n]} G_i$ and $\overline{\mathbf{g}}(\mathbf{x}^t) \triangleq \frac{1}{n} \sum_{i \in [n]} \mathbf{g}_i(\mathbf{x}^t)$. We have that,

$$\mathbb{E}\left[\|\mathbf{x}^{t+1} - \mathbf{x}^*\|_2^2 \big| \mathbf{x}^t\right] = \mathbb{E}\left[\|\mathbf{x}^t - \gamma_t \mathbf{g}(\mathbf{x}^t) - \mathbf{x}^*\|_2^2 \Big| \mathbf{x}^t\right]$$

$$= \|\mathbf{x}^t - \mathbf{x}^*\|_2^2 + \gamma_t^2 \mathbb{E}\left[\|\mathbf{g}(\mathbf{x}^t)\|_2^2 \Big| \mathbf{x}^t\right] - 2\gamma_t \mathbb{E}\left[\mathbf{g}(\mathbf{x}^t)^\top (\mathbf{x}^t - \mathbf{x}^*) \Big| \mathbf{x}^t\right]$$

$$= \|\mathbf{x}^t - \mathbf{x}^*\|_2^2 + \gamma_t^2 \mathbb{E}\left[\|\mathbf{g}(\mathbf{x}^t)\|_2^2 \Big| \mathbf{x}^t\right] - 2\gamma_t \mathbb{E}\left[\mathbf{g}(\mathbf{x}^t) \Big| \mathbf{x}^t\right]^\top (\mathbf{x}^t - \mathbf{x}^*)$$

$$\overset{(i)}{\leq} \|\mathbf{x}^t - \mathbf{x}^*\|_2^2 + \gamma_t^2 \mathbb{E}\left[\|\mathbf{g}(\mathbf{x}^t)\|_2^2 \Big| \mathbf{x}^t\right] - 2\gamma_t \left(f(\mathbf{x}^t) - f^*\right)$$

$$\overset{(ii)}{=} \|\mathbf{x}^t - \mathbf{x}^*\|_2^2 - 2\gamma_t \left(f(\mathbf{x}^t) - f^*\right)$$
$$+ \gamma_t^2 \left(\mathbb{E}\left[\|\mathbf{g}(\mathbf{x}^t) - \overline{\mathbf{g}}(\mathbf{x}^t)\|_2^2 \Big| \mathbf{x}^t\right] + \|\overline{\mathbf{g}}(\mathbf{x}^t)\|_2^2\right)$$

$$\overset{(iii)}{\leq} \|\mathbf{x}^t - \mathbf{x}^*\|_2^2 - 2\gamma_t \left(f(\mathbf{x}^t) - f^*\right) + \gamma_t^2 G_{\max}^2 \left(\frac{S(\mathbf{p}, \mathbf{P}, \mathbf{A})}{n^2} + 1\right) \tag{28}$$

In this chain of inequalities, the expectation is over the stochasticity of the intermittent links at iteration $t$. Here, (i) follows since we ensure that the stochastic subgradient computed at the parameter server despite the intermittent connectivity is unbiased, (ii) holds true for the same reason, and (iii) follows from Lipschitz continuity of $f_i$ for all $i \in [n]$ and the result on MSE for DME with collaborative relaying in the presence of intermittent connectivity. Now taking expectations over all sources of stochasticity, we have,

$$\mathbb{E}\left[\|\mathbf{x}^{t+1} - \mathbf{x}^*\|_2^2\right] \leq \mathbb{E}\left[\|\mathbf{x}^t - \mathbf{x}^*\|_2^2\right] - 2\gamma_t \left(\mathbb{E}[f(\mathbf{x}^t)] - f^*\right) + \gamma_t^2 G_{\max}^2 \left(\frac{S(\mathbf{p}, \mathbf{P}, \mathbf{A})}{n^2} + 1\right). \tag{29}$$

Summing all these inequalities for $t = 0, \ldots, T-1$, we get,

$$\mathbb{E}\left[\|\mathbf{x}^T - \mathbf{x}^*\|_2^2\right] \leq \|\mathbf{x}^0 - \mathbf{x}^*\|_2^2 - 2\sum_{t=0}^{T-1} \gamma_t \left(\mathbb{E}[f(\mathbf{x}^t)] - f^*\right) + G_{\max}^2 \left(\frac{S(\mathbf{p}, \mathbf{P}, \mathbf{A})}{n^2} + 1\right) \sum_{t=0}^{T-1} \gamma_t^2. \tag{30}$$

Since $\mathbb{E}\left[\|\mathbf{x}^T - \mathbf{x}^*\|_2^2\right] \geq 0$, this gives us,

$$\min_{t=0,\ldots,T-1} \mathbb{E}[f(\mathbf{x}^t)] - f^* \leq \frac{R^2 + G_{\max}^2 \left(\frac{S(\mathbf{p}, \mathbf{P}, \mathbf{A})}{n^2} + 1\right) \sum_{t=0}^{T-1} \gamma_t^2}{2 \sum_{t=0}^{T-1} \gamma_t}. \tag{31}$$

So, the algorithm converges to the optimal solution as long as the learning rate sequence $\{\gamma_t\}_{t \geq 0}$ is square-summable and not summable, i.e.

$$\gamma_t \geq 0, \quad \sum_{t=0}^{\infty} \gamma_t^2 < \infty, \quad \text{and} \quad \sum_{t=0}^{\infty} \gamma_t = \infty. \tag{32}$$

For a constant step-size sequence $\gamma_t = \gamma$ for all $t$, we have,

$$\min_{t=0,\ldots,T-1} \mathbb{E}[f(\mathbf{x}^t)] - f^* \leq \frac{R^2}{\gamma T} + \frac{\gamma G_{\max}^2}{2} \left(\frac{S(\mathbf{p}, \mathbf{P}, \mathbf{A})}{n^2} + 1\right) \tag{33}$$

So the algorithm converges to a neighborhood of the optimal solution at a convergence rate of $\mathcal{O}\left(\frac{1}{T}\right)$ and the size of the neighborhood depends on $S(\mathbf{p}, \mathbf{P}, \mathbf{A})$.

The results of this section can also be extended to setting with a stochastic subgradient oracle as shown in App. G.

# E   Proof of Theorem 4.2: Smooth and Polyak-Lojasiewicz (PL) Objectives

In this section, we derive the linear convergence rate of gradient descent with collaborative relaying (**ColRel-GD**) for smooth and PL objective functions. The algorithm is the same as Alg. 2 with each client evaluating the gradient $\nabla f_i(\mathbf{x}^t)$ at iteration $t$. We formally state the assumptions on the objective functions below.

**Assumption E.1.** (**Smoothness and Lower boundedness**). For every $i \in [n]$, the function $f_i$ is $L_i$-smooth, i.e., $\|\nabla f_i(\mathbf{x}) - \nabla f_i(\mathbf{y})\| \le L_i \|\mathbf{x} - \mathbf{y}\|_2$, for all $\mathbf{x}, \mathbf{y} \in \mathbb{R}^d$ and $f^* \triangleq \inf_{\mathbf{x} \in \mathbb{R}^d} f_i(\mathbf{x}) > -\infty$. As a consequence, $f = \frac{1}{n} \sum_{i \in [n]} f_i$ is L-smooth with $L \triangleq \frac{1}{n} \sum_{i \in [n]} L_i$.

**Assumption E.2.** (**Lipschitz continuity**). For $i \in [n]$, the local loss function $f_i$ is $G_i$-Lipschitz continuous, i.e., for any $\mathbf{x}, \mathbf{y} \in \mathbb{R}^d$, $|f_i(\mathbf{x}) - f_i(\mathbf{y})| \le G_i \|\mathbf{x} - \mathbf{y}\|_2$.

**Assumption E.3.** (**Polyak-Lojasiewicz (PL) condition**). For some $\mu > 0$, the global objective $f$ $\mu$-PL, i.e., for all $\mathbf{x} \in \mathbb{R}^d$, where $\mathbf{x}^* = \arg\min_{\mathbf{x} \in \mathbb{R}^d} f(\mathbf{x})$, we have, $f(\mathbf{x}) - f(\mathbf{x}^*) \le \frac{1}{2\mu} \|\nabla f(\mathbf{x})\|_2^2$.

We now state a descent lemma (without proof) from [25] that we use for our analysis.

**Lemma E.4** (Lemma 2, [25]). *Suppose that function $f$ is L-smooth and let $\mathbf{x}^{t+1} \leftarrow \mathbf{x}^t - \gamma \mathbf{g}^t$. Then for any $\mathbf{g}^t \in \mathbb{R}^d$ and $\gamma > 0$, we have,*

$$f(\mathbf{x}^{t+1}) \le f(\mathbf{x}^t) - \frac{\gamma}{2} \|\nabla f(\mathbf{x}^t)\|_2^2 - \left(\frac{1}{2\gamma} - \frac{L}{2}\right) \|\mathbf{x}^{t+1} - \mathbf{x}^t\|_2^2 + \frac{\gamma}{2} \|\mathbf{g}^t - \nabla f(\mathbf{x}^t)\|_2^2 \quad (34)$$

Conditioned on $\mathbf{x}^t$, the descent lemma as seen before, states that for a constant learning-rate $\gamma > 0$,

$$\mathbb{E}[f(\mathbf{x}^{t+1})|\mathbf{x}^t] - f(\mathbf{x}^*) \le \mathbb{E}[f(\mathbf{x}^t)|\mathbf{x}^t] - f(\mathbf{x}^*) - \frac{\gamma}{2} \|\nabla f(\mathbf{x}^t)\|_2^2$$

$$- \left(\frac{1}{2\gamma} - \frac{L}{2}\right) \mathbb{E}[\|\mathbf{x}^{t+1} - \mathbf{x}^t\|_2^2 | \mathbf{x}^t]$$

$$+ \frac{\gamma}{2} \mathbb{E}\left[\left\|\frac{1}{n} \sum_{i \in [n]} \tau_i^t \sum_{j \in [n]} \alpha_{ij} \tau_{ji}^t \nabla f_i(\mathbf{x}^t) - \frac{1}{n} \sum_{i \in [n]} \nabla f_i(\mathbf{x}^t)\right\|_2^2 \Big| \mathbf{x}^t \right]$$

$$\overset{(i)}{\le} \mathbb{E}[f(\mathbf{x}^t)|\mathbf{x}^t] - f(\mathbf{x}^*) - \frac{\gamma}{2} \|\nabla f(\mathbf{x}^t)\|_2^2 + \frac{\gamma G_{\max}^2}{2n^2} S(\mathbf{p}, \mathbf{P}, \mathbf{A}). \quad (35)$$

Here, the inequality (i) uses the fact that we choose our learning rate $\gamma \in \left(0, \frac{1}{L}\right]$ so that the term $\left(\frac{1}{2\gamma} - \frac{L}{2}\right)$ is non-negative, and the MSE result of DME (Theorem 3.2), as the last term can be upper bounded as,

$$\mathbb{E}\left[\left\|\mathbf{g}^t - \frac{1}{n} \sum_{i \in [n]} \nabla f_i(\mathbf{x}^t)\right\|_2\right] \overset{(ii)}{=} \mathbb{E}\left[\mathbb{E}\left[\left\|\mathbf{g}^t - \frac{1}{n} \sum_{i \in [n]} \nabla f_i(\mathbf{x}^t)\right\|_2 \Big| \mathbf{x}^t\right]\right] \overset{(iii)}{\le} \frac{G_{\max}^2}{n^2} \cdot S(\mathbf{p}, \mathbf{P}, \mathbf{A}). \quad (36)$$

Here, (ii) follows from the tower rule of conditional expectation, and the inner expectation is over the stochasticity of the intermittent connectivities, i.e., $\{\tau_i\}_{i \in [n]}$ and $\{\tau_{ij}\}_{i,j \in [n]}$. The inequality (iii) follows from the result on DME (Theorem 3.2), and $G_{\max} \triangleq \max_{i \in [n]} G_i$. Using the PL-inequality and taking expectation over all sources of stochasticity, we have,

$$\mathbb{E}[f(\mathbf{x}^{t+1})] - f(\mathbf{x}^*) \le (1 - \gamma\mu) \left(\mathbb{E}[f(\mathbf{x}^t)] - f(\mathbf{x}^*)\right) + \frac{\gamma G_{\max}^2}{2n^2} S(\mathbf{p}, \mathbf{P}, \mathbf{A}). \quad (37)$$

Recursively applying this for $t = 0, \ldots, T - 1$, we get,

$$\mathbb{E}[f(\mathbf{x}^T)] - f(\mathbf{x}^*) \leq (1 - \gamma\mu)^T \left(\mathbb{E}[f(\mathbf{x}^0)] - f(\mathbf{x}^*)\right) + \frac{\gamma G_{\max}^2}{2n^2} S(\mathbf{p}, \mathbf{P}, \mathbf{A}) \sum_{t=0}^{T-1} (1 - \gamma\mu)^t$$

$$\leq (1 - \gamma\mu)^T \left(\mathbb{E}[f(\mathbf{x}^0)] - f(\mathbf{x}^*)\right) + \frac{G_{\max}^2}{2n^2\mu} S(\mathbf{p}, \mathbf{P}, \mathbf{A}) \quad (38)$$

This shows that for smooth objective functions that satisfy the PL condition, ColRel converges to a neighborhood of the optimal solution and the size of the neighborhood is once again determined by $S(\mathbf{p}, \mathbf{P}, \mathbf{A})$. Note that the size of the convergence neighborhood goes to 0 as the number of clients $n \to \infty$. Furthermore, it is possible to get rid of the neighborhood and achieve exact convergence to the optimal solution using a diminishing learning-rate schedule. However, the convergence rate fails to be linear in that case, and deteriorates to a rate of $\mathcal{O}\left(\frac{1}{T}\right)$. Once again, results of Theorem 4.2 can be extended to the case with stochastic gradient oracle as discussed in App. G.

# F    Proof of Theorem 4.3: Smooth and Non-Convex Objectives

In this section, we derive the rate of convergence of gradient descent with collaborative relaying (**ColRel-GD**) to a stationary point for general non-convex objective functions. We still assume that the function is smooth and continuous as in Assumptions E.1 and E.2. The algorithm is the same as Alg. 2 with gradients $\nabla f_i(\mathbf{x}^t)$ evaluated at each client $i \in [n]$ instead of a subgradient. Using the general descent Lemma E.4,

$$\mathbb{E}f(\mathbf{x}^{t+1}) - f^* \leq \mathbb{E}f(\mathbf{x}^t) - f^* - \frac{\gamma}{2}\mathbb{E}\left[\|\nabla f(\mathbf{x}^t)\|_2^2\right]$$

$$- \left(\frac{1}{2\gamma} - \frac{L}{2}\right)\mathbb{E}\left[\|\mathbf{x}^{t+1} - \mathbf{x}^t\|_2^2\right] + \frac{\gamma}{2}\frac{G_{\max}^2}{n^2} \cdot S(\mathbf{p}, \mathbf{P}, \mathbf{A}). \quad (39)$$

Choosing the learning rate $\gamma \in \left(0, \frac{1}{L}\right]$, we get,

$$\mathbb{E}f(\mathbf{x}^{t+1}) - f^* \leq \mathbb{E}f(\mathbf{x}^t) - f^* - \frac{\gamma}{2}\mathbb{E}\left[\|\nabla f(\mathbf{x}^t)\|_2^2\right] + \frac{\gamma G_{\max}^2}{2n^2} \cdot S(\mathbf{p}, \mathbf{P}, \mathbf{A}). \quad (40)$$

Summing these inequalities for $t = 0, \ldots, T - 1$, we have,

$$\underbrace{\mathbb{E}f(\mathbf{x}^T) - f^*}_{D_T} \leq \underbrace{\mathbb{E}f(\mathbf{x}^t) - f^*}_{D_0} - \frac{\gamma}{2}\sum_{t=0}^{T-1}\mathbb{E}\left[\|\nabla f(\mathbf{x}^t)\|_2^2\right] + \frac{\gamma G_{\max}^2}{2n^2} \cdot S(\mathbf{p}, \mathbf{P}, \mathbf{A}) \cdot T \quad (41)$$

Since the expected suboptimality gap $D_T \geq 0$ and $D_0$ is the initial suboptimality gap. This yields,

$$\frac{1}{T}\sum_{t=0}^{T-1}\mathbb{E}\left[\|\nabla f(\mathbf{x}^t)\|_2^2\right] \leq \frac{2D_0}{\gamma T} + \frac{G_{\max}^2}{n^2} \cdot S(\mathbf{p}, \mathbf{P}, \mathbf{A}) \quad (42)$$

Note that the LHS of (42) can be interpreted as $\mathbb{E}\left[\|\nabla f(\widehat{\mathbf{x}})\|_2^2\right]$ where $\widehat{\mathbf{x}}$ is chosen uniformly at random from $\{\mathbf{x}_0, \ldots, \mathbf{x}^{T-1}\}$. Furthermore, (42) also implies that the best approximate stationary point amongst the iterates satisfies,

$$\min_{t=0,\ldots,T-1}\mathbb{E}\left[\|\nabla f(\mathbf{x}^t)\|_2^2\right] \leq \frac{2D_0}{\gamma T} + \frac{G_{\max}^2}{n^2} \cdot S(\mathbf{p}, \mathbf{P}, \mathbf{A}). \quad (43)$$

The convergence rate if $\mathcal{O}\left(\frac{1}{T}\right)$, and the iterates convergence to a neighborhood whose size is determined by $S(\mathbf{p}, \mathbf{P}, \mathbf{A})$. This is typical since in stochastic gradient descent, the iterates convergence to a neighborhood of a stationary point and the size of that neighborhood is determined by the variance of the stochastic gradients. In our setting, the stochasticity inherently arises from the intermittent links in the network, i.e., the TIV $\sigma_{\text{tv}}^2(\mathbf{p}, \mathbf{P}, \mathbf{A})$. Note that the size of the convergence neighborhood goes to 0 as the number of clients $n \to \infty$. This completes the proof.

Once again, the results of this section can be extended to the setting with stochastic gradient oracle, as discussed in App. G.

# G  Extensions to Stochastic (Sub) Gradient Oracle

In §3, we saw that intermittently connected links introduce an additional source of variance to the information received from each client at the PS. In principle, the TIV $\sigma_{\text{tv}}^2$ is similar to the variance due to other sources of stochasticity such as that arising from a stochastic (sub) gradient oracle. Since these sources of stochasticity are independent, the resulting variance at the PS is the sum of all the variances.

**Stochastic subgradient method with ColRel**. In the presence of a stochastic subgradient oracle, at any iteration $t$, each client $i \in [n]$ now returns $\widehat{\mathbf{g}}_i(\mathbf{x}^t)$. We consider the following standard assumptions on the subgradient oracle.[4]

1. **Unbiasedness**: For any $i \in [n]$ an $\mathbf{x}^t$, the stochastic subgradients satisfy, $\mathbb{E}[\widehat{\mathbf{g}}_i(\mathbf{x}^t)|\mathbf{x}^t] \in \partial f_i(\mathbf{x}^t)$.

2. **Bounded second moment**: There exists a constant $\widetilde{G} > 0$ such that for any $t$, the stochastic subgradients returned by the oracle satisfy, $\mathbb{E}[\|\widehat{\mathbf{g}}_i(\mathbf{x}^t)\|_2^2] \leq \widetilde{G}_i^2$.

A straightforward modification of the analysis in §D yields the same convergence rate as in Theorem 4.1 with $\widetilde{G}_{\max}$ instead of $G_{\max}$, where $\widetilde{G}_{\max} \triangleq \max_{i \in [n]} \widetilde{G}_i$.

**Stochastic gradient descent with ColRel**. Suppose at every iteration $t$, the stochastic gradient oracle at each client returns a stochastic gradient $\widehat{\mathbf{g}}_i(\mathbf{x}^t)$ that satisfies the following:

1. **Unbiasedness**: For all $i \in [n]$ and any iteration $t$, we have, $\mathbb{E}\left[\widehat{\mathbf{g}}_i(\mathbf{x}^t)\right] = \nabla f_i(\mathbf{x}^t)$.

2. **Bounded second moment**: For all $i \in [n]$ and any iteration $t$, we have, $\mathbb{E}\left[\|\widehat{\mathbf{g}}_i(\mathbf{x}^t)\|_2^2\right] \leq \beta \|\nabla f_i(\mathbf{x}^t)\|_2^2 + \sigma^2$, where $\beta$ and $\sigma^2$ are non-negative constants and depend inversely with the batch size used for computing the stochastic gradient.

With minimal modification in the analysis that we derive in App. E, one can show that the following holds for smooth and PL objectives,

$$\mathbb{E}[f(\mathbf{x}^T)] - f(\mathbf{x}^*) \leq (1 - \gamma\mu)^T \left(\mathbb{E}[f(\mathbf{x}^0)] - f(\mathbf{x}^*)\right) + \frac{(\beta G_{\max}^2 + \sigma^2)\sigma_{\text{tv}}^2}{2\mu}. \tag{44}$$

Whereas, for smooth non-convex objectives, we have,

$$\min_{i=0,\ldots,T-1} \mathbb{E}\left[\|\nabla f(\mathbf{x}^t)\|_2^2\right] \leq \frac{2D_0}{\gamma T} + (\beta G_{\max}^2 + \sigma^2)\sigma_{\text{tv}}^2. \tag{45}$$

# H  Numerical Simulations: Extended

In this section, we first present numerical simulation results on DME in §H.1, and then we discuss the federated optimization simulations of §5 in more detail.

## H.1  Distributed Mean Estimation (DME)

For the simulations in this section, we have considered a cluster of clients that communicate with the PS to estimate the mean of their vectors. Amongst the clients, a few "good" clients have a high probability of successfully transmitting their data to the PS, while the remaining "bad" clients can only convey their data to the PS by relaying through one of the good clients. The number of good clients is varied along the $X$-axis while the MSE for estimating the mean is plotted on the $Y$-axis.

In Fig. 3, we plot and visually compare naïve and ColRel strategies. The naïve strategy uses no collaborations, i.e., we set $p_{ij} = 0$ for all $i \neq j$, with $\mathbf{A} = \text{diag}(p_1^{-1}, \ldots, p_n^{-1})$. Although without collaboration, this choice of $\mathbf{A}$ for the naïve strategy ensures an unbiased estimate at the PS (i.e. (2) is satisfied), albeit with a higher MSE. We consider $n = 10$ clients and the dimension $d = 100$, with the probability of a good client successfully transmitting to the PS is $p_{\text{good}} = 0.5, 0.7, 0.9$ (varied across the different plots), whereas the bad clients have a probability $p_{\text{bad}} = 0.2$. The clients can collaborate

---

[4]These assumptions can be relaxed with a more rigorous analysis. However, for simplicity, we do not do so here as that is not the major focus of our work.

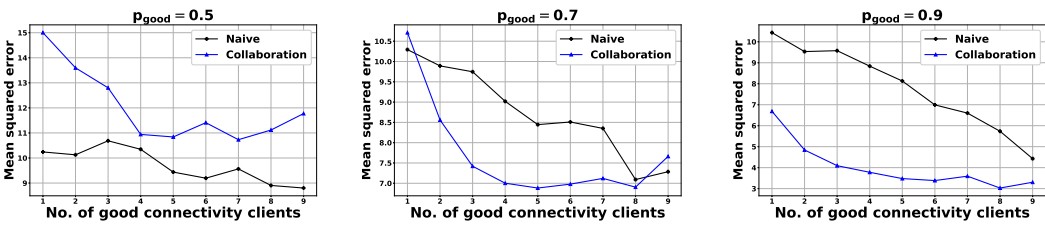

Figure 3: Erdős-Rényi topology for collaboration with $p_c = 0.8$, dimension $d = 100$.

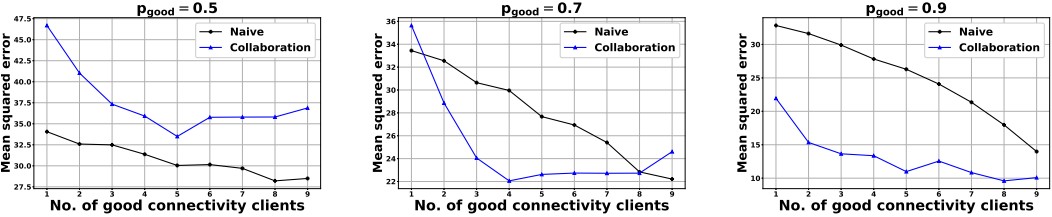

Figure 4: Erdős-Rényi topology for collaboration with $p_c = 0.8$, dimension $d = 1000$.

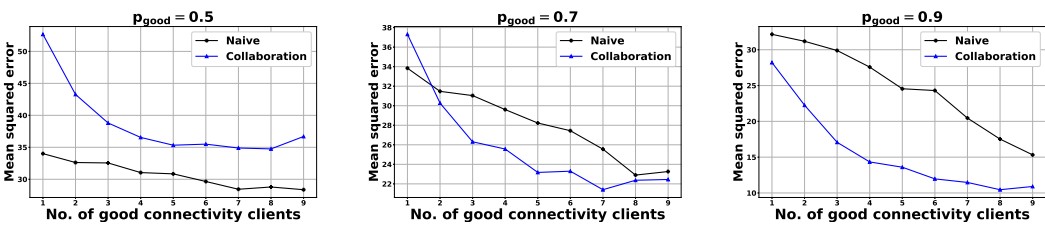

Figure 5: Ring topology with each client having 6 neighbors with $p_c = 0.8$, dimension $d = 1000$.

with each other over a fully connected topology with intermittent client-client connections that are successful with probability $p_c = 0.8$. That is, the collaboration is done over an Erdős-Rényi graph, $G(n = 10, p = 0.8)$. The data at each client is generated from a Gaussian distribution $\mathcal{N}(0, 1)$, and then each coordinate is raised to the power of 3. Raising the coordinates of a Gaussian vector to the third power generates a heavy-tailed distribution. Consequently, if a client that has vector with a few large coordinate values is unable to convey its data to the PS due to a failed transmission (as is the case in naïve DME), this can incur a significant MSE. On the contrary, ColRel ensures that the poor connectivity clients can still relay their updates to neighboring nodes, resulting in comparatively smaller MSE. The plot are averaged over 50 realizations. The data and the intermittent connectivity is generated independently at each realization. While generating the intermittent connectivity between clients, we ensure that if client-$i$ does not collaborate with client-$j$, then even client-$j$ does not collaborate with client-$i$. For instance, this is motivated by the setting where a physical obstacle blocks communication between two clients over a wireless network.

In Fig. 4, we increase the dimension to $d = 1000$ and obtain the same plots. In Fig. 5, we consider a different topology for client-client collaboration, wherein clients are constrained to communicate with other clients over a ring topology, i.e., $p_{ii} = 1$ for all $i \in [n]$, $p_{ij} = p_c = 0.8$ iff $j = i\pm1, i\pm2, i\pm3 \mod n$, and $p_{ij} = 0$ otherwise. In the next subsections §H.2, we discuss our federated learning simulation setup in more details.

## H.2 Federated Learning Simulations

**Simulation setup**: We train a ResNet-20 model for image classification on the CIFAR-10 [22] dataset. We distributed the training set of $50,000$ images across $n = 10$ clients in both independent and identically distributed (IID) and non-IID fashions. Non-IIDness of the data distribution amongst clients is prevalent in FL setups and to emulate it, we consider the *sort-and-partition* approach wherein the training data is initially sorted based on the labels, and then they are divided into blocks and distributed among the clients randomly based on a parameter $s$, that measures the skewness of

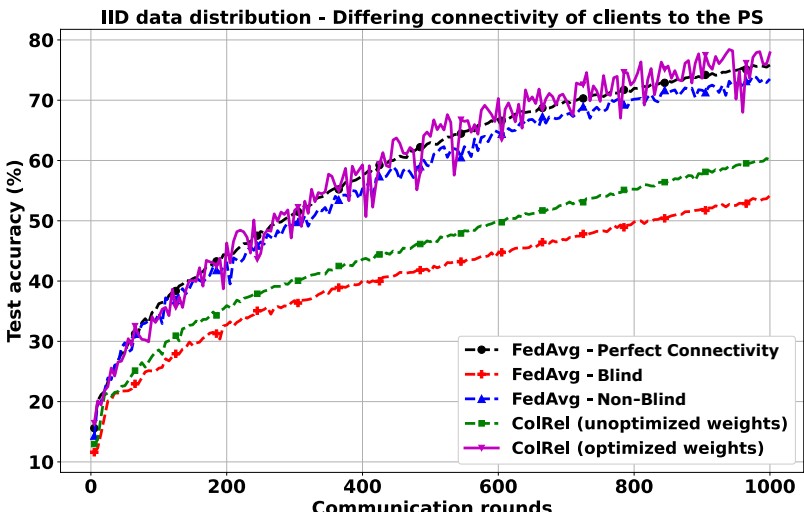

Figure 6: Ring topology with 2 neighbors per client

the data distribution. More precisely, $s$ defines the maximum number of different labels present in the local dataset of each user, and therefore, smaller $s$ implies more skew in the data distribution. We use $s = 3$, i.e. each client has images from at most 3 classes.

The plotted results of all simulations are averaged over 5 independent realizations. In between every communication round to the parameter server (PS), the clients execute 8 local training steps of local-SGD. We utilize the SGD optimizer at the clients with a global momentum ($\beta = 0.9$) at the PS, a learning rate of $0.05$ for SGD, a coefficient of $10^{-4}$ for $\ell_2$-regularization to prevent overfitting, and a batch-size of $64$. All simulations were carried out on NVIDIA GeForce GTX $1080$ Ti with a CUDA Version $11.4$. We also note that we have ensured all the simulations to have the same step-size, i.e., the learning rate for different simulations has not been tuned individually.

We consider the following benchmarks with which we compare our proposed ColRel scheme:

1. **FedAvg – perfect connectivity**. We consider FedAvg when all clients are able to successfully transmit their local updates to the PS at every communication round. This serves as a natural upper bound to the performance of any algorithm proposed in the presence of intermittent connectivity.

2. **FedAvg – Blind**. As a natural performance lower bound in the presence of intermittent client connectivity, we consider a naïve FedAvg strategy wherein the PS is unaware of the identity of clients. In this strategy, for the clients that are unable to send their updates to the PS due to a communication failure, the PS simply assumes that their update is zero. Essentially, the PS adds all the local updates it receives at any communication round, and divides it by the total number of clients irrespective of the knowledge of the number of actual successful transmissions. Such blind averaging strategies are often the norm for FEEL employing OAC.

3. **FedAvg – Non-Blind**. As another benchmark, we also consider a non-blind strategy, where the PS is aware of the identity of the clients, and knows exactly, how many and which clients have successfully been able to send their local update to the PS. This is common in point-to-point learning settings. In this case, the PS simply ignores the clients that have been unable to send their updates, and averages the successful updates by dividing the global aggregate at the PS by the number of successful transmissions.

In our simulations in Figs. 2a, 2b, and 2c (reproduced here as Figs. 6, 7, and 8 for better resolution), we compare the above-mentioned benchmarks with our proposed ColRel strategy in the presence of intermittent connectivity of clients to the PS, as well as amongst themselves. In order to demonstrate the improved performance of our proposed strategy with respect to the above-mentioned benchmarks, we consider the following setups:

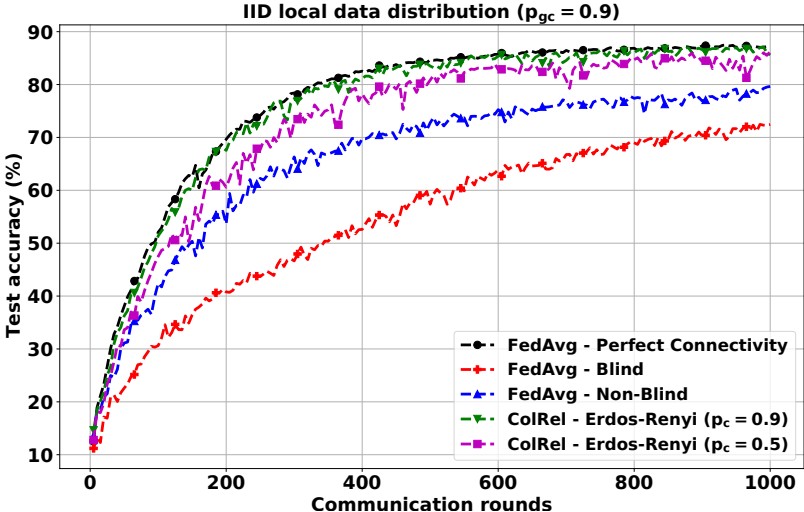

Figure 7: Single good connectivity client with an Erdős-Rényi graph

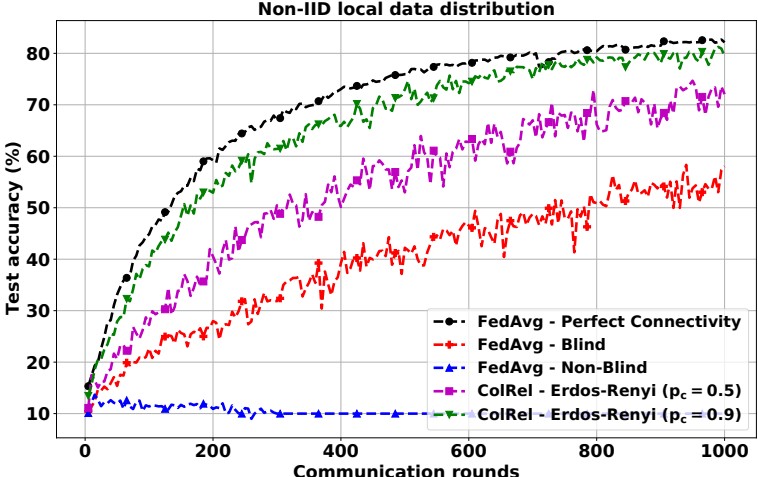

Figure 8: SGD with global momentum with an Erdős-Rényi graph

1. **Effect of topology**: The decentralized topology according to which the clients collaborate in sharing their updates plays an important role in the performance of ColRel. In Fig. 2a, we consider a ring topology in which each client is constrained to communicate with only 2 of their neighboring clients. In this setting, when a connection is present between any two clients (so that it presents an opportunity for collaboration), the communication link is *not* intermittent. More specifically, the matrix $\mathbf{P}$ is given by, $p_{ij} = 1$ iff $j = -1, 0, or, +1$ mod $n$. On the other hand, in Fig. 2b, we consider client-client collaboration over an Erdős-Rényi topology In other words, each client can collaborate with any other client but over intermittently connected links, i.e., $0 < p_{ij} < 1$ for $i \neq j$. Although over intermittent links, this provides an increased opportunity for collaborations, and consequently, the final test accuracy (after 1000 communication rounds) is higher in Fig. 2b compared to Fig. 2a.

Furthermore, later in this section, we also present a brief discussion for FL over networks with mmWave links, where the connectivity probabilities are determined by the clients' and PS's geographical locations. In particular, we investigate a setting where intermittent connectivity between clients might be preferred over perfect connectivity because it leads to more increased collaboration on average.

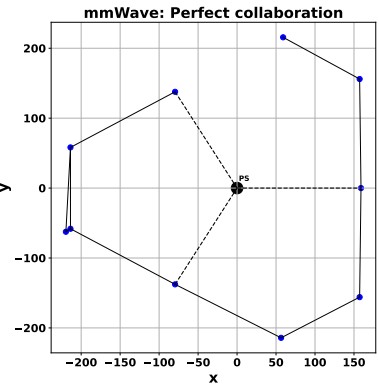

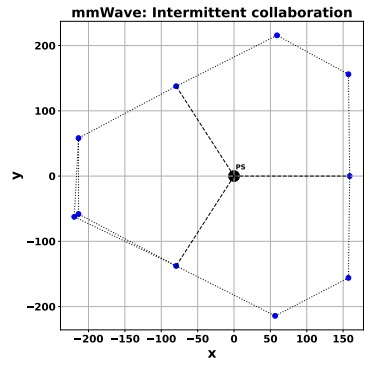

(a) Perfect client-client collaboration

(b) Intermittent client-client collaboration

Figure 9: Network topologies for perfect vs. intermittent client-client connectivities.

2. **Non-IID data distribution**: We evaluate the performance of COLREL in the presence of non-IID local data distribution. COLREL outperforms other strategies when some clients with important local data have persistently poor connectivity.

    In Figs. 2a an 2b, we consider an iid data distribution amongst clients. In this case, FedAvg – Non-Blind performs quite comparably to ColRel. This is because in the absence of any non-iidness in the local data distribution amongst clients, a naïve strategy like simply ignoring the clients with failed transmissions, and averaging only the received updates still performs reasonably well, because the aggregated gradient at the PS is still an unbiased estimate of the true global gradient at the PS, albeit with a slightly higher variance due to a fewer number of clients that successfully transmit.

    However, such a naïve strategy fails in Fig. 2c where we consider a non-iid local data distribution across clients. This is because an important subset of the training data is being possessed by clients that have poor connectivity to the PS and cannot relay their updates to the PS. ColRel ensures that even the clients with bad connectivity to the PS can relay their updates, ensures a convergence rate comparable to FedAvg - perfect connectivity.

3. **Heterogeneity in client connectivity**: Different clients may have different connectivity to the PS, and we show that as long as there is one client with reasonably good connection to the PS, its neighbors can relay their updates to it – yielding improved performance for COLREL. We consider this in all the simulation plots we have simulated.

**Network topology with mmWave links**: We now consider a network topology with mmWave links in which the probability of successful transmissions between clients and between clients and the PS are determined according to [2] as a function of distance ($\delta$) between them as, $p = \min\left(1, e^{-\delta/30+5.2}\right)$.

Here, we consider a comparison between non-intermittent (perfectly available) client-client collaboration, vs. intermittent client-client collaboration. This is similar to the difference in topologies of Fig. 2a (where the client-client collaboration is deterministic), versus Fig. 2b (which considers intermittency that we have allowed for in the analytical formulation as well).

In Fig. 9a, we consider that two clients can collaborate perfectly if $p_{ij} \geq 0.99$ and determine the network topology accordingly. In other words, $p_{ij}$ and $p_{ji}$ are high enough to consider the links to be consistently present. The PS is at the origin and the clients are distributed in a way such that only three of them can communicate with the PS. Similarly, in Fig. 9b, the connectivities between clients are intermittent. To avoid links that are too unreliable, if $p_{ij} < 0.5$, we consider those clients do not collaborate. The geographical locations of the clients is the same in both the scenarios. Note that Fig. 9b has a few additional links compared to Fig. 9a. In Fig. 10, we note that allowing for intermittent collaboration amongst clients results in improved convergence rate of training. Moreover, both perfect as well as intermittent connectivity outperform FedAvg without collaboration.

**Network topology with mmWave links for n = 20 clients**: We increase the number of clients in our mmWave system from 10 to 20. In this setup, we also reduced the non-iid skewness parameter of local data distribution to $s = 2$. We consider two topologies (hereafter referred to as Setup 1

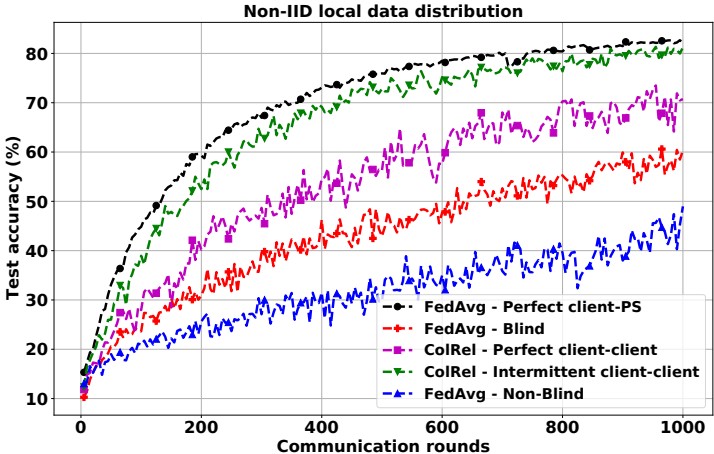

Figure 10: Improved convergence due to increased collaboration with intermittent client-client connectivity

and Setup 2) for geographic locations of the clients. Setups 1 and 2 are visualized in Figs. 11 and 13 respectively; clients in Fig. 11 enjoy better connectivity to the PS and to one another. In these figures red lines depict client-client connections, and black lines denote client-PS connections. Additionally, solid lines denote stable connections where the connectivity probability is 1, and dashed and dotted lines denote intermittently connected links. As before, the connectivity probabilities are determined as characterized by mmWave links [2], according to $p = \min\left(1, \mathrm{e}^{-\delta/30+5.2}\right)$, where $\delta$ is the distance between two nodes. In Setup 1, we have the following probabilities of intermittently connected client-client links: $p_{ij} = 0.8$ where the unordered pairs $\{i, j\}$ can take the values $\{1, 4\}, \{2, 3\}, \{6, 9\}, \{7, 8\}, \{12, 13\}, \{14, 17\}, \{16, 19\}, \{17, 18\}$. In Setup 2, we have the following probabilities of intermittently connected client-client links: $p_{ij} = 0.93$ where the unordered pairs $\{i, j\}$ can take the values $\{2, 3\}, \{7, 8\}, \{12, 13\}, \{17, 18\}$. Furthermore, the client-PS connectivity probabilities for Setups 1 and 2 are

$$\mathbf{p} = [0.8, 0.03, 0.03, 0, 0.01, 0.8, 0.03, 0.03, 0, 0.01,$$
$$0.8, 0.03, 0.03, 0, 0.01, 0.8, 0.03, 0.03, 0, 0.01],$$

and

$$\mathbf{p} = [0.92, 0.05, 0.04, 0.01, 0.11, 0.92, 0.05, 0.04, 0.01, 0.11,$$
$$0.92, 0.05, 0.04, 0.01, 0.11, 0.92, 0.05, 0.04, 0.01, 0.11],$$

respectively.

We plot the resulting test accuracy as a function of the number of rounds for Setups 1 and 2 in Figs. 12 and 14, respectively. The numerical results for each of these figures were averaged over 5 realizations. Additionally, we compare our proposed ColRel algorithm with the FedAvg - perfect connectivity, FedAvg - Blind and FedAvg - Non-Blind. The performance improvement of our proposed ColRel algorithm is consistent as seen in Figs. 12 and 14; in Setup 1, ColRel has approximately 40% higher accuracy with respect to FedAvg - Blind and 68% with respect to FedAvg - Non-Blind. Furthermore, a comparison between Figs. 12 and 14 shows that increasing the connectivity of client-client links and client-PS lead to a better final accuracy. We note that while the accuracy of all the setups with intermittently connected links increased as the connectivity improves, our proposed ColRel algorithm still outperforms the FedAvg - Blind and FedAvg - Non-Blind algorithms by a large margin. In fact for Setup 2, the ColRel algorithm leads to a training loss that is comparable to the FedAvg without any intermittent connectivity, i.e., FedAvg - Perfect Connectivity.

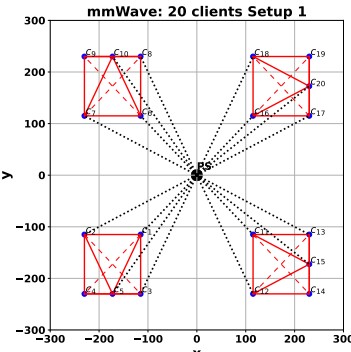

Figure 11: Client and PS locations in a mmWave system for **Setup 1**.

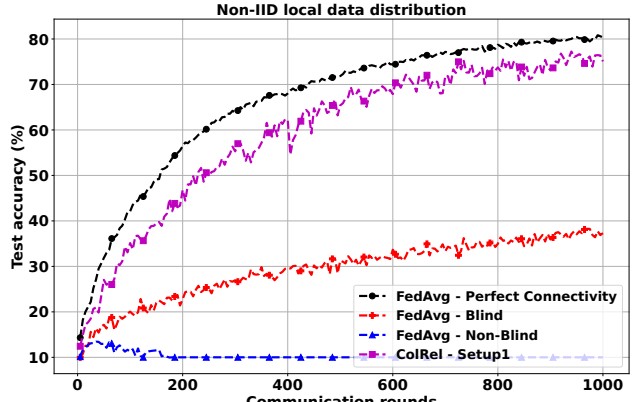

Figure 12: Comparison of algorithms for **Setup 1**

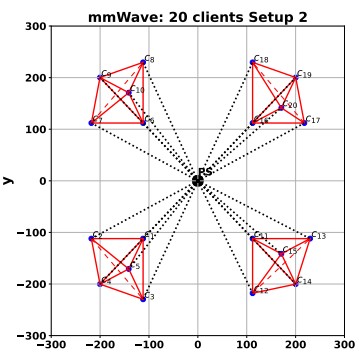

Figure 13: Client and PS locations in a mmWave system for **Setup 2**.

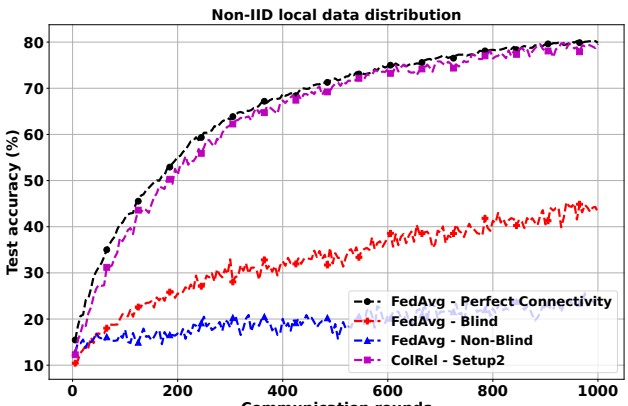

Figure 14: Comparison of algorithms for **Setup 2**

