# OpenReview forum: "ColRel: Collaborative Relaying for Federated Learning over Intermittently Connected Networks"
_NeurIPS.cc/2022/Workshop/Federated_Learning — FL-NeurIPS 2022 Poster_

### Official Review · Reviewer_gPj4 · 2022-10-11
**Collaborative Relaying can improve FL systems when clients have intermittent connectivity**

This paper proposes a collaborative relaying (ColRel) system that aims to improve the convergence of FL systems when clients have intermittent connectivity to the central server. Clients participating in the relaying system exchange and aggregate model updates with neighbours before sending them to the central server. When clients have more reliable connections to neighbours than the central server, as may be the case in IoT networks, ColRel can more reliably include information from all clients.

Quality:
The work presents a clear motivation and method, with convergence analysis for some types of loss functions, and some experimental justification.

Clarity:
The paper is well written and presented in a logical manner.

Originality:
To my knowledge the collaborative relaying framework and convergence analysis is original.

Significance:
The ideas will be interesting to the FL research community.

Pros:
 - When clients have more reliable connections to neighbours than to the central server, their model updates can be more consistently included.
 - The mathematical framework is laid out clearly, and the theorems are interesting (I have not verified the proofs).

Cons:
 - The privacy of data is reduced. Model updates are widely shared among clients rather than a single (trusted) central server.
 - The overall communication used is increased by a potentially large multiple.
 - Clients which are already struggling to communicate their updates are now required to communicate their updates many times, adding resource pressure.
 - Clients must receive and aggregate multiple model updates which significantly increases the computational burden (especially memory) on edge devices which are likely to be resource constrained.

I would encourage the authors to address the increase in resources quantitatively in the future, and to think about how the additional privacy exposure of model updates to (potentially unfriendly) neighbours could be mitigated, for example with differential privacy or proxy models [Shen et al. "Federated Mutual Learning"; Kalra et al. "ProxyFL: Decentralized Federated Learning through Proxy Model Sharing"].

I think the convex relaxation in eq. 5 should be further justified and discussed both theoretically and empirically. It also seems unrealistic that the probabilities $p_i$ and $p_{ij}$ would be known in practice, which is a prerequisite for finding $\alpha_{ij}$. It would be good to discuss and experimentally show how these probabilities can be determined when a network starts up, and how they can be updated as the connectivities evolve online.

Possible typo in line 122 $P(\tau_{ij} | \tau_{ij})$.

---

> ### Author Response · Authors · 2022-11-26
> **Response to Reviewer gPj4**
>
> Thank you for your review. We appreciate your valuable time and feedback.
>
> >The privacy of data is reduced. Model updates are widely shared among clients rather than a single (trusted) central server.
>
> We agree with you that privacy is an important aspect that needs to be considered in collaborative FL systems. This is the subject of our ongoing research. More specifically, clients can add local perturbations to ensure local differential privacy. Together with random client participation due to the intermittent nature of the communications to the parameter server and between themselves, it is possible to derive privacy guarantees.
>
> > The overall communication used is increased by a potentially large multiple. Clients which are already struggling to communicate their updates are now required to communicate their updates many times, adding resource pressure.
>
> This may or may not be the case, depending on the system implementation. For instance, in wireless sensor networks, the transmit power available at any edge device determines the probability of a successful transmission. If an edge device has less transmit power available so that it cannot send its data directly to the server, it can still possibly send its data to a neighboring client. To do this, it can simply broadcast, and all neighbors in close proximity will receive it. The neighborhood in this case is determined by the range allowed by the transmit power available; and there is just a single transmission with multiple receivers (broadcast channel). This does not necessarily increase the number of communications.
>
> > Clients must receive and aggregate multiple model updates which significantly increases the computational burden (especially memory) on edge devices which are likely to be resource constrained.
>
> This also, may or may not be the case, depending on the actual system implementation. In this work, we have focused primarily on communication stragglers, i.e., edge devices that cannot send their updates to the server due to a lack of communication opportunity, for instance, due to a physical obstacle. The client may not necessarily be computation constrained in this case.
>
> That being said, the only added computational burden is due to the fact that each client now has to compute a weighted average of its neighbors' update. Since in many settings, the number of neighbors each client communicates with (on an average) is often small, this added computation is not likely to be a significant bottleneck, especially since clients are able to compute gradients (of the same dimension) on mini batches of data stored on them. We agree with you that a more detailed discussion on this aspect is definitely going to be helpful.
>
> > I think the convex relaxation in eq. 5 should be further justified and discussed both theoretically and empirically.
>
> We propose to minimize the convex relaxation and later fine tune the solution of this to arrive at a stationary point of the original optimization problem (4), primarily because the optimization landscape of (4) is not straightforward to obtain. In many cases the gap between (4), and the convex relaxation (5), is quite small. A careful analysis of the proof of Thm. 3.2 shows that from eq. (14), when the cross products between the vectors at different nodes is 0, the original optimization problem (4) is convex and a relaxation is not necessary. Consequently, when these cross products are small, the gap due to relaxation is small. Numerical experiments validate this and we will discuss it in the final archival version of this work too.
>
> > It also seems unrealistic that the probabilities $p_i$ and $p_{ij}$ would be known in practice, which is a prerequisite for finding $\alpha_{ij}$. It would be good to discuss and experimentally show how these probabilities can be determined when a network starts up, and how they can be updated as the connectivities evolve online.
>
> This is an interesting point, and we agree with you that these probabilities can be estimated online as well, so that the empirically estimated probabilities converge to the true probabilities, and subsequently, the collaboration weights converge to the optimal weights as the algorithm proceeds.
>
> Nevertheless, our assumption of knowing $p_i$ and $p_{ij}$ beforehand is not unrealistic, because in wireless systems these transmission probabilities can easily be estimated using pilot signals, provided that the time scale over which these probabilities change is longer than the time scale over which the algorithm executes. Moreover, for some wireless channels, these probabilities can be estimated using empirical models -- for instance, in Sec. H.2 (Appendix), we consider simulations where these connections are mmWave links, wherein these probabilities are functions of the distance between the transmitter and the receiver.
>
> Thank you again for pointing out the typo which we have corrected.

---

### Official Review · Reviewer_5GuY · 2022-10-16
**Review for "ColRel: Collaborative Relaying for Federated Learning over Intermittently Connected Networks"**

The paper proposes an interesting, seemingly novel strategy for making FL more robust in edge devices, where each client sends not only its own update but an aggregate of neighboring clients' updates. It is well-written; nicely motivated, providing a good explanation of why such a strategy is needed; and results are presented clearly.

The strategy is derived by looking at the distributed mean estimation problem, and its consequences to convex optimizations are then assessed. Numerical experiments are performed to show how it can be useful in different settings.

A discussion around privacy considerations is missing. Could for instance the clients perform secure aggregation? This is an important point in federated learning and should not be ignored.

Overall, it seems like the proposed algorithm can be quite useful in edge scenarios, even though its practical implementation can be tricky (how to have clients securely communicating etc.).

Minor points:
- the way ColRel oscillates in the numerical experiments should be better explained
- it is not clear to me what is precisely the difference between blind and non-blind FedAvg; how come the blind approach does not fail in the non-iid case?

---

> ### Author Response · Authors · 2022-11-26
> **Response to Reviewer 5GuY**
>
> Thank you for your review. We appreciate your valuable time and feedback.
>
> We looked at the problem of mean estimation primarily for introducing the notion of topology-induced variance due to the stochastic nature of intermittent connections, so that it can be integrated as a black box with existing FL systems.
>
> We agree with you that privacy is an important aspect that needs to be considered in collaborative FL systems. This is the subject of our ongoing research. More specifically, clients can add local perturbations to ensure local differential privacy. Together with random client participation due to the intermittent nature of communications to the parameter server and between themselves, it is possible to derive privacy guarantees.
>
> > the way ColRel oscillates in the numerical experiments should be better explained
>
> This is an important point and we will study this in more detail in our future works. Our current hypothesis is that oscillations are more when some of the transmission probabilities are small, i.e., close to $0$. Then, in order to ensure unbiasedness, some of the weights are larger, as the unbiasedness condition $\sum_{i \in [n]}p_j p_{ij} \alpha_{ji} = 1$, suggests that the weights and probabilities are inversely related. Since the *effective step size* in which the global iterate takes a descent direction is larger with larger $\alpha$, this leads to more oscillation. For instance, in Fig. 7 (appendix), the green plot (with corresponds to a higher degree of connectivity, as it is an Erdos-Renyi graph with client-client connectivity, $p_c = 0.9$) exhibits fewer oscillations that the purple plot ($p_c = 0.5$).
>
> > it is not clear to me what is precisely the difference between blind and non-blind FedAvg; how come the blind approach does not fail in the non-iid case?
>
> Blind FedAvg refers to the aggregation rule, $\frac{1}{n}\sum_{i=1}^{n}\tau_i\mathbf{x}_i$, whereas, non-blind averaging refers to the rule,
>
> $\frac{1}{(\sum_{i \in [n]}\tau_i)}\sum_{i=1}^{n}\tau_i\mathbf{x}_i$. Our hypothesis as to why the blind approach does not fail in the non-iid case is as follows -- Since the denominator in the non-blind case is a random variable, it can be close to $0$, especially when clients have small values of $p_i$. In such a case, the *effective step size* could get large, leading to divergence of the algorithm. On the other hand, the denominator in the blind case is fixed, i.e., $n$, and consequently, the step size is small, which can sometimes lead to a slow convergence, rather than divergence. Nevertheless, it is important to note that blind and non-blind FedAvg are just obvious benchmarks in the presence of intermittent connectivity that we compared ColRel with, and their behavior has not been rigorously analyzed to make any concrete claims.

---

### Official Review · Reviewer_Axxj · 2022-10-17
**A new way to estimate the average over all clients under intermittent connection**

Pros.

1. This paper propose a way to estimate the average over all clients under intermittent connection. Convergence upper bounds are derived for the proposed communication pattern.

Cons.

1. The proposed ColRel is very similar to that in [6] so it is not new. The only difference is that clients with stable connection work like a local server.

2.  The paper seems not considering multiple local updates, which is the most important feature of federated learning compared with distributed learning.

3. Lemma 3.1 says the chosen $\alpha$ should bring an unbiased estimation, which is strong.

Q1. Can any given $p_i$ and $p_{ij}$ allow to find such an $\alpha$? what is the restriction for $p_i$ and $p_{ij}$?

Q2. If the number of local updates is large, how about this condition and how to minimize the variance?

Summary.

This paper proposed a way to optimize $\alpha$ such that with an unbiased estimation for the averaged gradients, the variance is minimized. However, this paper does not consider multiple local updates. The framework is not new compared to previous works.

---

> ### Author Response · Authors · 2022-11-26
> **Response to Reviewer Axxj**
>
> Thank you for your review. We appreciate your valuable time and feedback.
>
> > The proposed ColRel is very similar to that in [6] so it is not new. The only difference is that clients with stable connection work like a local server.
>
> Our work and [6] have several differences, some of which we mentioned below:
>
> 1. The work [6] does not consider intermittently connected networks. The connections are always present, or always absent, as a result of which [6] can determine which of the nodes can act as local servers, which is time invariant. On the contrary, in our work, each node can act as a relay for the updates of its neighboring nodes, and even the connections between the nodes for local collaboration can be intermittent.
>
> 2. The work [6] considers disjoint subnetworks, each of which communicates with a local hub. Our setup does not consider such a constraint, and in some sense, can be viewed as a generalization of [6], since now, one particular client can send its update to more than one hubs.
>
> A core contribution of our work is to analyze stochastic networks and determine collaboration weights as a function of the probabilities of links between client-PS or client-client being present. The resulting water-filling type algorithm described in Appendix C gives optimal choice of weights so that the updates are relayed more to clients better connectivity, but the connectivity need not be perfect, i.e., it need not be always present.
>
> > The paper seems not considering multiple local updates, which is the most important feature of federated learning compared with distributed learning. If the number of local updates is large, how about this condition and how to minimize the variance?
>
> It is not too different to consider multiple local updates at the clients too. One of our primary goals in this work is to show the effect of intermittently connected networks, and consequently, the role of topology induced variance. The topology induced variance will still be an additional source of variance even if multiple local updates were present. Please refer to the arxiv version of our paper: https://arxiv.org/abs/2202.11850 which considers multiple local updates too, and is under review at an archival venue.
>
> We agree with you that considering multiple local updates is important, and it might be possible to derive a modified weight optimization scheme that jointly takes into account this aspect as well.
>
> > Lemma 3.1 says the chosen $\alpha$ should bring an unbiased estimation, which is strong.
>
> We agree with you that it is not a necessary condition to ensure convergence of collaborative relaying. But it is a sufficient condition that ensures equal importance is given to the data from each client. It might be possible to relax this assumption if we are aware of some side information such as correlations between the data across different clients. Relaxing such a constraint would require us to appropriately modify the weight optimization procedure.
>
> > Can any given $p_i$ and $p_{ij}$ allow to find such an $\alpha$? What is the restriction for $p_i$ and $p_{ij}$?
>
> The feasible set of the proposed optimization problem is non-empty, i.e., for any given $p_i$ and $p_{ij}$, it is always possible to find weights $\{\alpha_{ij}\}_{i,j \in [n]}$.
>
> This can be seen because the trivial solution in which clients do not collaborate with their neighbors is always a feasible solution, i.e., we set $\alpha_{ii} = \frac{1}{p_i}$ for all $i \in [n]$, and set $\alpha_{ij} = 0$ for all $i \neq j$. Solving the optimization problem (4) can only lead to a smaller topology induced variance.

---

### Decision · Program_Chairs · 2022-10-20

Accept (Poster)